# LIGHT-X: GENERATIVE 4D VIDEO RENDERING WITH CAMERA AND ILLUMINATION CONTROL

**Tianqi Liu**[1,2,3]**, Zhaoxi Chen**[1]**, Zihao Huang**[1,2,3]**, Shaocong Xu**[2]**, Saining Zhang**[2,4]**,**
**Chongjie Ye**[5]**, Bohan Li**[6,7]**, Zhiguo Cao**[3]**, Wei Li**[1]**, Hao Zhao**[4,2†]**, Ziwei Liu**[1†]

[1]S-Lab, Nanyang Technological University     [2]Beijing Academy of Artificial Intelligence
[3]AIA, Huazhong University of Science and Technology     [4]AIR, Tsinghua University
[5]FNii, The Chinese University of Hong Kong, Shenzhen     [6]Shanghai Jiao Tong University
[7]Ningbo Institute of Digital Twin, Eastern Institute of Technology, China

`https://lightx-ai.github.io/`

Figure 1: **Light-X** enables controllable video relighting and redirection from monocular video inputs, supporting illumination editing guided by either background images (**top**) or text prompts (**middle**), as well as camera trajectory redirection with user-defined trajectories (**bottom**).

## ABSTRACT

Recent advances in illumination control extend image-based methods to video, yet still facing a trade-off between lighting fidelity and temporal consistency. Moving beyond relighting, a key step toward generative modeling of real-world scenes is the joint control of camera trajectory and illumination, since visual dynamics are inherently shaped by both geometry and lighting. To this end, we present **Light-X**, a video generation framework that enables controllable rendering from monocular videos with both viewpoint and illumination control. **1)** We propose a disentangled design that decouples geometry and lighting signals: geometry and motion are captured via dynamic point clouds projected along user-defined camera trajectories, while illumination cues are provided by a relit frame consistently projected into the same geometry. These explicit, fine-grained cues enable effective disentanglement and guide high-quality illumination. **2)** To address the lack of paired multi-view and multi-illumination videos, we introduce **Light-Syn**, a degradation-based pipeline with inverse-mapping that synthesizes training pairs from in-the-wild monocular footage. This strategy yields a dataset covering static, dynamic, and AI-generated

---

† Corresponding authors.

scenes, ensuring robust training. Extensive experiments show that Light-X outperforms baseline methods in joint camera-illumination control and surpasses prior video relighting methods under both text- and background-conditioned settings.

# 1 INTRODUCTION

Real-world scenes are inherently rich, dynamic, and high-dimensional, shaped jointly by geometry, motion, and illumination. Yet monocular videos, the dominant medium for capturing everyday life, record only a 2D projection of this complexity. Unlocking controllable video generation with camera and illumination control would allow us to revisit such footage from novel viewpoints and under diverse lighting, thereby enabling immersive AR/VR experiences and flexible filmmaking pipelines.

Progress toward this goal has evolved along two largely independent lines of research: video relighting and camera-controlled video generation. **In the relighting domain**, existing video relighting methods typically extend single-image pipelines such as IC-Light (Zhang et al., 2025b) to the video setting, either through training-free fusion (Zhou et al., 2025) or by introducing architectural modifications (Fang et al., 2025). But they suffer from a fundamental trade-off between lighting fidelity and temporal coherence, and crucially, they do not support camera control. **On the other hand**, camera-controlled video generation approaches (YU et al., 2025; Bai et al., 2025; Zhang et al., 2025a; Liu et al., 2025a) enable novel-view video synthesis with accurate camera motion and strong spatio-temporal consistency. However, they are limited to viewpoint manipulation and lack the ability to edit illumination, leaving the joint control of lighting and camera trajectory an open challenge.

In this paper, we aim to develop a video generation model that jointly controls camera trajectory and illumination from monocular videos. This goal raises two key challenges: **1) Joint control.** Controlling camera trajectory and illumination together is inherently difficult, as it demands disentangled yet coherent modeling of geometry, motion, and lighting. Even for video relighting alone, existing methods struggle to balance lighting fidelity with temporal consistency. Viewpoint changes exacerbate this trade-off, making joint camera–illumination control especially challenging. **2) Data scarcity.** Training requires paired multi-view and multi-illumination videos to disentangle geometry and lighting, but such data are unavailable in real-world settings.

To address these challenges, we propose the following solutions. **1) Disentangled control formulation.** We introduce a conditioning scheme that explicitly decouples geometry/motion from illumination. Camera trajectories are modeled through dynamic point cloud rendering like (YU et al., 2025), while illumination cues are provided by projecting a relit frame (obtained via (Zhang et al., 2025b)) into the same geometry, so that the model simultaneously receives projected original frames for geometry and motion, and a projected relit frame for lighting. These fine-grained cues greatly facilitate model learning. In addition, we introduce a light-DiT layer that enforces global illumination consistency. **2) Degradation-based data curation.** Since paired multi-view and multi-illumination videos are scarce, we design Light-Syn, a degradation-based pipeline with inverse mapping that synthesizes training pairs from in-the-wild footage. Degraded video variants (*e.g.*, relit or edited) serve as inputs, while the originals provide supervision[1]. By applying the inverse mapping of the degradation process, we project geometry and lighting cues from the original video into the degraded view, yielding diverse pairs from AI-generated, static, and dynamic scenes for robust generalization.

Building on these foundations, we present Light-X, the first framework for video generation with joint control of camera and illumination from monocular videos. As shown in Fig. 1, by decoupling camera and lighting conditioning, our method supports joint camera–illumination control, video relighting, and novel-view synthesis within a single model. Extensive evaluations demonstrate that our approach consistently outperforms baselines in joint camera–illumination control (Table 1). For individual tasks, it delivers superior lighting fidelity and temporal consistency in video relighting under both text (Table 3) and background conditions (Table 5). In addition, a soft-weighted illumination mask enables seamless integration of diverse lighting cues, such as environment maps and reference images, further improving the flexibility.

In summary, **1)** We propose Light-X, the first framework for video generation with joint control of camera trajectory and illumination from monocular videos. **2)** We develop Light-Syn, a degradation-based data pipeline with inverse geometric mapping, which constructs paired training data under

---

[1]As method outputs are typically lower in fidelity than natural footage, we refer to them as *degraded*.

controlled camera viewpoints and lighting. **3)** We introduce a disentangled conditioning scheme that explicitly separates geometry and motion from illumination cues, enabling both independent and coupled control. **4)** Extensive experiments show that Light-X achieves SOTA performance in joint camera–illumination control and video relighting under text- and background-conditioned settings.

## 2 RELATED WORK

**Video Generative Models** have progressed from GANs (Goodfellow et al., 2020; Clark et al., 2019; Tulyakov et al., 2018; Vondrick et al., 2016) and VAEs (Kingma & Welling, 2013; Kalchbrenner et al., 2017; Mathieu et al., 2015; Ranzato et al., 2014) to autoregressive transformers (Wu et al., 2022). Then research focuses shifted to diffusion models (Ho et al., 2020). VDM (Ho et al., 2022) first used a 3D U-Net for video synthesis, and Make-A-Video (Singer et al., 2023) improved resolution and frame rate via super-resolution and interpolation. Latent diffusion (Rombach et al., 2022) was later adopted for efficiency (Blattmann et al., 2023; Zhou et al., 2022; He et al., 2022; Xing et al., 2023; Chen et al., 2024b; Guo et al., 2023b; Wang et al., 2024b). Most recently, Sora (Brooks et al., 2024) demonstrated the scalability of Diffusion Transformers (DiT) (Peebles & Xie, 2023), inspiring many DiT-based models (Wan et al., 2025; Yang et al., 2024b; Kong et al., 2024; Fan et al., 2025; Ma et al., 2025; Lin et al., 2024). Building on these advances, we leverage video diffusion priors for controllable video synthesis.

**Learning-Based Illumination Control** enables manipulation of scene lighting in images or videos. Early studies leveraged physical illumination models (Barron & Malik, 2014) or deep networks with explicit lighting representations (Zhou et al., 2019; Sun et al., 2019), especially for portraits (Shu et al., 2017; Shih et al., 2014; Sengupta et al., 2018; Chen & Liu, 2022). The recent success of diffusion models has greatly advanced relighting fidelity (Cha et al., 2025; Jin et al., 2024; Kim et al., 2024; Zhang et al., 2025b; He et al., 2025; Liang et al., 2025; Chaturvedi et al., 2025; Chadebec et al., 2025). IC-Light (Zhang et al., 2025b) employs a light-transport consistency loss with large-scale datasets to achieve high-quality image relighting. Recent works (Zhou et al., 2025; Fang et al., 2025; Zeng et al., 2025; Lin et al., 2025) have extended image relighting to videos. Light-A-Video (Zhou et al., 2025) employs cross-frame light attention and progressive fusion in a training-free manner, while RelightVid (Fang et al., 2025) extends IC-Light's 2D U-Net to a 3D backbone with temporal attention. Yet, these methods still struggle with the trade-off between lighting fidelity and temporal consistency. In this paper, we propose a unified video generation framework that achieves temporally consistent and high-fidelity video relighting, while also supporting camera control.

**Camera-Controlled Video Generation.** Recent advances in video generation have emphasized conditional signals for controllable synthesis (Yin et al., 2023; Guo et al., 2023a; Xing et al., 2024; Fu et al., 2024). Camera-controlled methods (Yang et al., 2024a; Bahmani et al., 2024; Zheng et al., 2024; 2025) integrate camera parameters into diffusion models for viewpoint control. In static scenes, pose-conditioned diffusion has evolved from object-level (Liu et al., 2023c;d; Long et al., 2023; Tang et al., 2024; Chen et al., 2025c; Ye et al., 2025) to scene-level methods (Gao et al., 2024; Ren et al., 2025; Sargent et al., 2023; Liu et al., 2024; Yu et al., 2024; Chen et al., 2025a). For dynamic settings, some methods (Wang et al., 2024c; He et al., 2024; Sun et al., 2024; Xiao et al., 2024; Bai et al., 2025; Li et al., 2025a) exploit camera parameters or trajectories for novel-view videos, while others (Wu et al., 2025; Kuang et al., 2024; Bian et al., 2025; Liu et al., 2025a; Wang et al., 2025a) develop multi-view video diffusion. Another direction (You et al., 2025; Gu et al., 2025; Zhang et al., 2025a; YU et al., 2025) leverages explicit geometric cues such as depth or tracking to guide camera control. However, existing methods remain focused solely on camera trajectory control. We instead pursue joint control of camera motion and illumination for high-quality, controllable video generation.

## 3 METHOD

Given a monocular source video $\boldsymbol{V}^s = \{\boldsymbol{I}_i^s\}_{i=1}^f$, our objective is to synthesize a target video $\boldsymbol{V}^t = \{\boldsymbol{I}_i^t\}_{i=1}^f$ of the same dynamic scene, but re-rendered under user-specified camera trajectories and illumination conditions. The camera trajectory is denoted as $\mathcal{C} := \{[\boldsymbol{R}_i, \boldsymbol{t}_i] \in \mathbb{R}^{3 \times 4}\}_{i=1}^f$, where $\boldsymbol{R}_i$ and $\boldsymbol{t}_i$ represent the rotation and translation of the $i$-th frame relative to the original coordinate system. The illumination condition is denoted as $\mathcal{L}$, which may be provided in various forms (*e.g.*, a text prompt, an HDR environment map, or a reference image) and will be discussed later. The

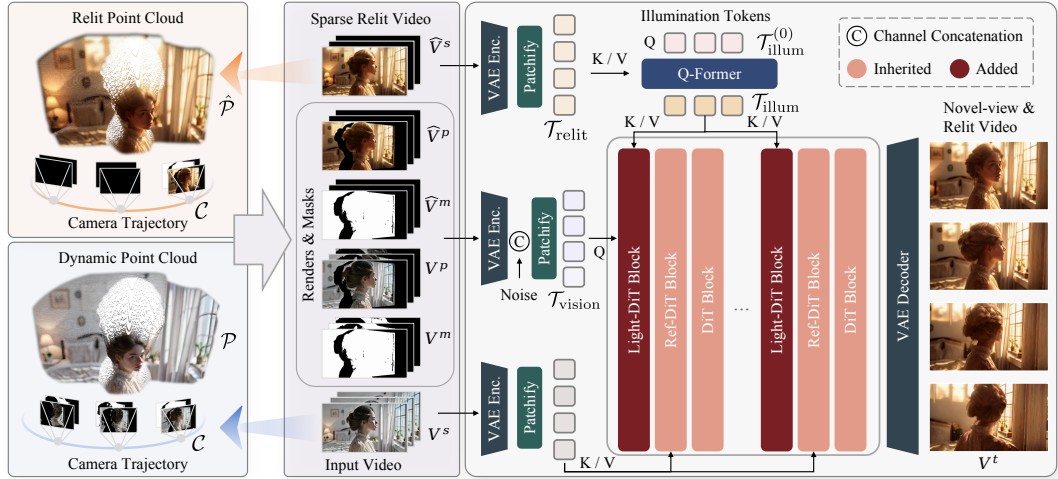

Figure 2: Overview of **Light-X**. Given an input video $V^s$, we first relight one frame with (Zhang et al., 2025b), conditioned on a lighting text prompt, to obtain a sparse relit video $\hat{V}^s$. We then estimate depths to construct a dynamic point cloud $\mathcal{P}$ from $V^s$ and a relit point cloud $\hat{\mathcal{P}}$ from $\hat{V}^s$. Both point clouds are projected along a user-specified camera trajectory, producing geometry-aligned renders and masks $(V^p, V^m)$ and $(\hat{V}^p, \hat{V}^m)$. These six cues, together with illumination tokens extracted via a Q-Former, are fed into DiT blocks for conditional denoising. Finally, a VAE decoder reconstructs a high-fidelity video $V^t$ faithful to the target trajectory and illumination.

generated video $V^t$ should faithfully preserve the appearance and dynamics of $V^s$ while adhering to $\mathcal{C}$ and $\mathcal{L}$. In the following sections, we first introduce the camera–illumination decoupling strategy (Sec. 3.1), then present the camera–illumination conditioned video diffusion model (Sec. 3.2). We next describe the data curation pipeline Light-Syn (Sec. 3.3) and finally discuss the framework's flexibility under diverse illumination conditions (Sec. 3.4).

## 3.1 FORMULATION: CAMERA–ILLUMINATION DECOUPLING

As illustrated in Fig. 2, given a input source video $V^s$, we disentangle camera and illumination control by constructing two point clouds that separately encode geometric and lighting information.

**Camera control.** To accurately regulate the camera trajectory, inspired by (Yu et al., 2024; YU et al., 2025; Guo et al., 2025; Hu et al., 2025), we leverage dynamic point clouds as an explicit inductive bias for modeling viewpoint transformations. Concretely, we first estimate a sequence of depth maps $D^s = \{D_i^s\}_{i=1}^f$ from the source video $V^s$ using video depth estimation approaches (Hu et al., 2024). Each frame is then back-projected to 3D space to form a dynamic point cloud $\mathcal{P} = \{P_i\}_{i=1}^f$:

$$P_i = \Phi^{-1}(I_i^s, D_i^s; K), \tag{1}$$

where $\Phi^{-1}$ denotes the inverse perspective projection and $K \in \mathbb{R}^{3\times3}$ is the camera intrinsics matrix. Given a user-specified trajectory $\mathcal{C} = \{[R_i, t_i]\}_{i=1}^f$, the point clouds are projected into the target viewpoints, yielding geometry-aligned views $V^p = \{I_i^p\}_{i=1}^f$ and visibility masks $V^m = \{M_i^p\}_{i=1}^f$:

$$I_i^p, M_i^p = \Phi(R_i P_i + t_i; K). \tag{2}$$

Together, these projected views and their masks serve as a strong geometric prior, guiding the diffusion model to produce videos that remain geometrically coherent along the specified trajectory.

**Illumination control.** For illumination, we apply IC-Light (Zhang et al., 2025b) to an arbitrary frame from the source video (the first frame is used for illustration in Fig. 2), producing an image relit according to the desired textual prompt. Subsequently, we construct a sparse relit video $\hat{V}^s = \{\hat{I}_i^s\}_{i=1}^f$, in which the relit frame is retained while all other frames remain blank. Using the previously estimated depths $\{D_i^s\}$, together with the camera intrinsics $K$ and extrinsics $\{[R_i, t_i]\}$, this sparse

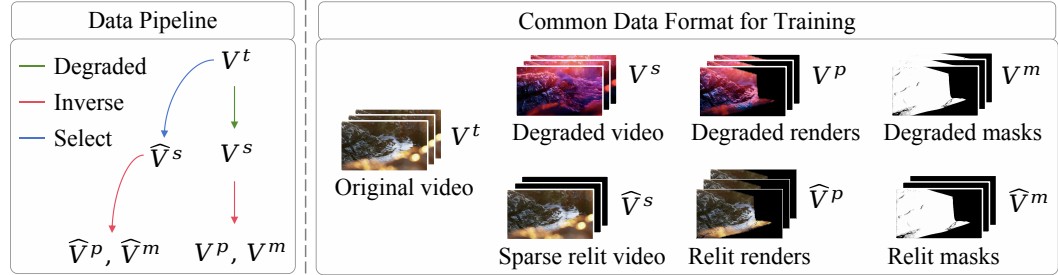

Figure 3: Overview of **Light-Syn**. From an in-the-wild video $V^t$, we generate a degraded $V^s$ and derive renders, masks $(V^p, V^m)$, and relit counterparts $(\hat{V}^p, \hat{V}^m)$ via inverse transformations.

relit video is lifted into a relit point cloud $\hat{\mathcal{P}} = \{\hat{P}_i\}_{i=1}^f$:

$$\hat{P}_i = \Phi^{-1}(\hat{I}_i^s, D_i^s; K). \tag{3}$$

We reuse the depths predicted from the original video, rather than estimating them again from the relit video, to ensure geometric alignment between the relit and original content. Analogous to the source video branch, the relit point cloud is projected along the target trajectory, yielding geometrically aligned relit views $\hat{V}^p = \{\hat{I}_i^p\}_{i=1}^f$ with corresponding binary masks $\hat{V}^m = \{\hat{M}_i^p\}_{i=1}^f$, which indicate where illumination information is available and serve as lighting cues:

$$\hat{I}_i^p, \ \hat{M}_i^p = \Phi(R_i\hat{P}_i + t_i; \ K). \tag{4}$$

## 3.2 Architecture: Camera–Illumination Conditioned Video Diffusion

With the obtained projected source views $V^p$ and masks $V^m$, together with the relit projections $\hat{V}^p$ and masks $\hat{V}^m$, the target video can be formulated as a conditional distribution as

$$x \sim p\big(x \mid V^s, \hat{V}^s, V^p, \hat{V}^p, V^m, \hat{V}^m\big), \tag{5}$$

which not only provides explicit geometric and illumination cues, but also disentangles the two factors in a geometrically aligned space, offering fine-grained guidance and enabling effective learning.

**Fine-grained cues.** The conditional cues $V^p$, $V^m$, $\hat{V}^p$, and $\hat{V}^m$ are first fed into the VAE encoder. The resulting latents are concatenated with sampled noise (see the middle of Fig. 2) along the channel dimension and then patchified into a sequence of vision tokens $\mathcal{T}_{\text{vision}}$. These tokens encode two complementary fine-grained cues: the projected views $V^p$, which carry scene content, geometry, and motion, and the projected relit views $\hat{V}^p$, which provide illumination cues. These tokens are then merged along the sequence axis with text tokens $\mathcal{T}_{\text{text}}$ (not shown in Fig. 2 due to space limit) obtained from the source video via (Li et al., 2022) and (Raffel et al., 2020). The fused text-vision tokens are then passed through DiT blocks for denoising.

**Global control.** While the rendered fine-grained cues facilitate learning of camera and illumination control, we observe that illumination strength gradually diminishes as the synthesized frames move further away from the relit frame. To mitigate this issue, we introduce a global illumination control module. Specifically, we encode the relit frame with a VAE encoder and patchify its latent to obtain the relit token $\mathcal{T}_{\text{relit}}$. Inspired by (Liu et al., 2023a; Xing et al., 2023), we employ a Q-Former (Li et al., 2023) to extract illumination information. A set of learnable illumination tokens $\mathcal{T}_{\text{illum}}^{(0)}$ serves as queries, while the relit token $\mathcal{T}_{\text{relit}}$ provides the keys and values (Fig. 2 right top). The resulting tokens $\mathcal{T}_{\text{illum}}$ are then injected into our introduced Light-DiT layer through cross-attention:

$$\mathcal{T}_{\text{vision}}' = \text{CrossAttn}\Big(Q = \mathcal{T}_{\text{vision}}, K = V = \mathcal{T}_{\text{illum}}\Big) \tag{6}$$

In addition, we retain the original DiT and Ref-DiT modules from (YU et al., 2025), which respectively aggregate text-vision information and preserve 4D consistency with the input source video.

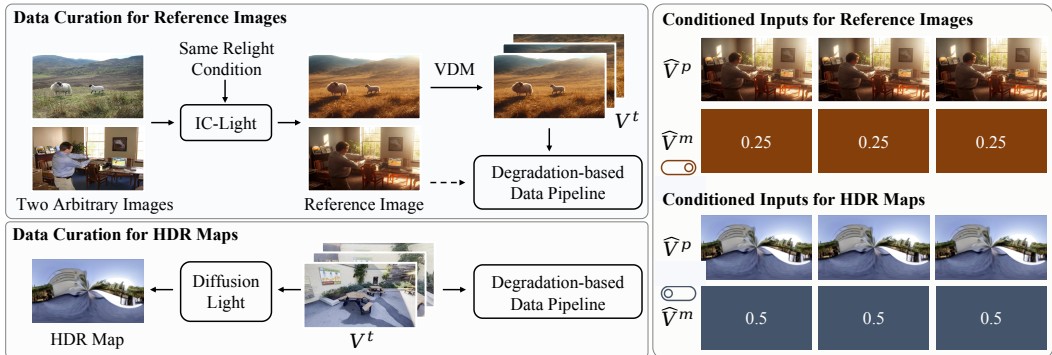

Figure 4: Left: Data curation pipelines for reference-image and HDR-map conditioned video generation. Right: Conditioning cues with soft masks used for model training.

## 3.3 DATA CURATION

Effective training requires paired videos with varied camera viewpoints and illumination, yet collecting such data in the real world is almost infeasible. We first analyze the training data requirements in detail and then introduce Light-Syn, a degradation-based pipeline with inverse mapping for synthesizing paired data from in-the-wild monocular videos, as illustrated in Fig. 3.

**Training Data Requirements.** Our model takes as input a source video $V^s$, a target video $V^t$, and conditioning sequences $V^p$, $\hat{V}^s$, and $\hat{V}^p$, each with specific requirements. The target $V^t$ should have high quality and consistency, and the input $V^s$ must remain 4D-consistent with $V^t$ in overlapping regions. Projected source views $V^p$ provide reliable geometric priors aligned with the target, while the sparse relit video $\hat{V}^s$ provides explicit illumination cues matching the target's lighting, and the projected relit views $\hat{V}^p$ deliver fine-grained lighting information geometrically aligned with $V^p$.

**Light-Syn Pipeline.** To construct such training pairs, we take an in-the-wild video as target $V^t$, degrade it to obtain $V^s$, and record the degradation transformations. Applying their inverses transfers the geometry and illumination of $V^t$ onto $V^s$, producing spatially aligned conditioning cues. We curate our dataset from three complementary sources: static scenes (8k), dynamic scenes (8k), and AI-generated videos (2k), where the first provide accurate multi-view data, the second capture realistic motion, and the third enrich illumination diversity. All sources satisfy the training requirements, yielding paired inputs, targets, and geometrically aligned conditioning cues. Detailed construction procedures are illustrated in Fig. C and described in detail in Sec. B of the Appendix.

## 3.4 FRAMEWORK FLEXIBILITY

**Camera–illumination decoupled control.** Although our training data are curated for joint control, the decoupling and masking mechanisms also allow for independent usage flexibly. For camera control, the conditioned relit frame is replaced with the original frame to preserve lighting. For illumination control (*i.e.*, video relighting), we set $V^p = V^s$, make $V^m$ fully visible, and substitute $\hat{V}^p$ with the sparse relit video $\hat{V}^s$, with $\hat{V}^m$ updated accordingly. Our framework also supports foreground video relighting conditioned on background images, where $V^s$ is composed of the foreground video and background using foreground masks, and the sparse relit video $\hat{V}^s$ is generated with IC-Light (Zhang et al., 2025b). Further details are provided in Sec. A.2 of the Appendix.

**Extension to diverse illumination conditions.** Our framework has the potential to accommodate diverse illumination hints as conditioning inputs, such as HDR environment maps and reference images. A reference image here denotes an image from a different scene that conveys lighting information, analogous to a style-transfer source. As shown in Fig. 4, we extend the data curation pipeline accordingly. For HDR maps, we extract environment lighting with DiffusionLight (Phongthawee et al., 2024) and apply the degradation pipeline to obtain 16k samples from (Wang et al., 2023; Ling et al., 2024). For reference-image conditioning, we generate pairs of IC-Light (Zhang et al., 2025b) relit images with matched prompts (Lin et al., 2014; Team, 2024): one serves as the

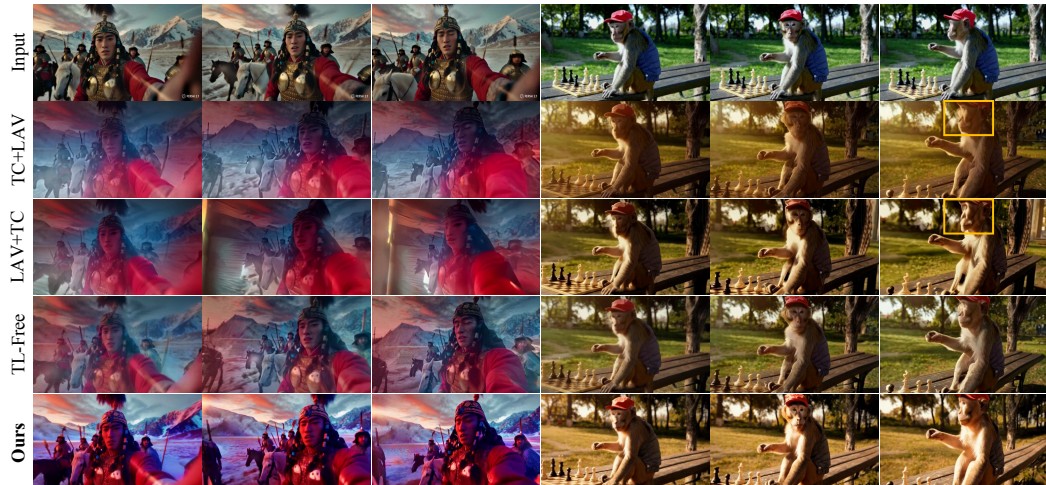

Figure 5: Qualitative comparison for camera-illumination control with light prompts "neon light" (left) and "sunlight" (right). Our method outperforms baselines in relighting quality, temporal consistency, and novel-view content generation. Refer to the supplementary video for clearer comparisons.

Table 1: Quantitative results for the joint camera-illumination control task. User preference indicates the percentage of participants who selected our method.

| Method | Image Quality | | Video Smoothness | | User Study (%, Ours) | | | | Time ↓ |
|---|---|---|---|---|---|---|---|---|---|
| | FID ↓ | Aesthetic ↑ | Motion Pres. ↓ | CLIP ↑ | RQ | VS | IP | 4DC | |
| TC+IC-Light | / | 0.573 | 6.558 | 0.976 | 89.3 | 91.7 | 88.3 | 88.5 | 3.25 min |
| TC+LAV | 138.89 | 0.574 | 4.327 | 0.986 | 86.0 | 84.4 | 88.0 | 89.0 | 4.33 min |
| LAV+TC | 144.61 | 0.596 | 5.027 | 0.987 | 85.1 | 89.3 | 88.8 | 87.5 | 4.33 min |
| TL-Free | 122.73 | 0.595 | 3.356 | 0.987 | 88.0 | 89.2 | 88.2 | 88.2 | 5.50 min |
| Ours | **101.06** | **0.623** | **2.007** | **0.989** | / | / | / | / | 1.83 min |

illumination reference, while the other is animated into $V^t$ by a commercial video model (Team, 2024), yielding about 1k samples for each of the text- and background-conditioned settings. During training, conditioning inputs are assigned by modality:

$$(\hat{V}^p, \hat{V}^m) = (V_k, \alpha_k \mathbf{1}), \quad k \in \{\text{ref}, \text{hdr}\}, \tag{7}$$

with $\alpha_{\text{ref}} = 0.25$ and $\alpha_{\text{hdr}} = 0.50$. These soft masks act as domain indicators (Chen et al., 2025b), enabling a single model to generalize across diverse illumination conditions.

## 4 EXPERIMENTS

### 4.1 EXPERIMENTAL SETTINGS

**Baselines.** Our evaluation focuses on two tasks: joint camera-illumination control and video relighting. For the joint control, as no prior work addresses it directly, we construct baselines by combining existing methods: TrajectoryCrafter (TC) (YU et al., 2025)+IC-Light (Zhang et al., 2025b), Light-A-Video (LAV) (Zhou et al., 2025)+TC, TC+LAV, and a training-free baseline TL-Free (Sec. A.1). For video relighting, we assess both text- and background-conditioned settings, comparing with IC-Light (Zhang et al., 2025b), IC-Light+AnyV2V (Ku et al., 2024), Light-A-Video (Zhou et al., 2025), and RelightVid (Fang et al., 2025). We use the Wan2.1 (Wan et al., 2025) implementation of LAV with default hyperparameters. As RelightVid currently offers only a background-conditioned model, we evaluate it exclusively in that setting. More details are provided in Sec. A.3 of the Appendix.

**Metrics.** Following (Zhou et al., 2025), the evaluation focuses on two aspects: relighting quality and temporal consistency. Relighting quality is measured by FID (Heusel et al., 2017) between each method's outputs and frame-wise IC-Light results, and by the Aesthetic Preference metric, defined as the mean of the aesthetic score and image quality in (Huang et al., 2024). Temporal consistency

Table 2: Evaluation of joint camera–illumination control using real in-the-wild videos as reference.

| Method | PSNR ↑ | SSIM ↑ | LPIPS ↓ | FVD ↓ |
|---|---|---|---|---|
| TC + IC-Light | 10.96 | 0.456 | 0.474 | 58.85 |
| TC + LAV | 12.18 | 0.470 | 0.508 | 73.78 |
| LAV + TC | 12.48 | 0.463 | 0.479 | 60.95 |
| TL-Free | 13.49 | 0.547 | 0.418 | 54.44 |
| Ours | **13.96** | **0.582** | **0.378** | **45.91** |

Table 3: Quantitative results for video relighting. $^{*}$ indicates evaluation on the first 16 frames.

| Method | Image Quality | | Video Smoothness | | User Study (%, Ours) | | | Time ↓ |
|---|---|---|---|---|---|---|---|---|
| | FID ↓ | Aesthetic ↑ | Motion Pres. ↓ | CLIP ↑ | RQ | VS | IP | |
| IC-Light | / | 0.632 | 3.293 | 0.983 | 88.3 | 90.3 | 91.7 | 1.42 min |
| LAV | 112.45 | 0.614 | 2.115 | 0.991 | 85.2 | 88.5 | 92.5 | 2.50 min |
| Ours | **83.65** | **0.645** | **1.137** | **0.993** | / | / | / | 1.50 min |
| IC-Light+AnyV2V | 106.05 | 0.612 | 3.777 | 0.985 | 97.6 | 95.1 | 98.4 | 1.67 min |
| Ours$^{*}$ | **77.97** | **0.625** | **1.452** | **0.992** | / | / | / | / |

is assessed through the average CLIP (Radford et al., 2021) similarity between consecutive frames and Motion Preservation, computed as the deviation between RAFT (Teed & Deng, 2020) estimated optical flow and that of the source video. Considering that IC-Light-referenced FID may induce a bias toward the reference model, we additionally perform an evaluation that compares model outputs directly against real in-the-wild videos. In this evaluation protocol, real videos are treated as ground truth. Their lighting descriptions are extracted using LLaVA (Liu et al., 2023b), and a degraded counterpart is synthesized using LAV (Zhou et al., 2025) under a neutral-lighting prompt to serve as the model input. At test time, the degraded video is paired with its LLaVA-inferred lighting prompt, which is provided as the illumination condition. The relit outputs are then assessed against the real videos using standard perceptual and temporal metrics, including PSNR, SSIM (Wang et al., 2004), LPIPS (Zhang et al., 2018), and FVD (Unterthiner et al., 2019). We also conduct a user study with 57 participants to evaluate relighting quality (RQ, lighting fidelity and alignment with the prompt), video smoothness (VS), ID preservation (IP, consistency of the object's identity and albedo after relighting), and 4D consistency (4DC, spatio-temporal coherence in the novel-view setting). During evaluation, lighting prompts, directions, and camera trajectories are randomly sampled for each video.

**Datasets.** For evaluation data, we collect 200 high-quality videos from sources including Pexels (Pexels, 2025), Sora (Brooks et al., 2024), and Kling (Team, 2024). These videos cover a wide range of subjects such as humans, animals, and objects, across diverse scenes with substantial motion, incorporating both in-the-wild and AI-generated content. For background conditioned relighting, we use 10 background images and 30 foreground videos from (Zhang et al., 2025b) and (Team, 2024), producing 300 combinations. None of these videos is used for training in any compared methods.

**Implementation Details.** The framework is based on (YU et al., 2025; CogVideoX-Fun, 2024; Yang et al., 2024b). Training uses videos of resolution $384 \times 672$ and 49 frames, for 16,000 iterations with a learning rate of $2 \times 10^{-5}$ and a batch size of 8 on eight H100 GPUs. Video depths are estimated with (Hu et al., 2024) to construct dynamic point clouds, with camera intrinsics set empirically.

## 4.2 CAMERA-ILLUMINATION CONTROL RESULTS

Qualitative results are shown in Fig. 5 and better inspected in supplementary videos. TC (YU et al., 2025)+LAV (Zhou et al., 2025) is limited by LAV's weak relighting, especially under large camera motion, causing poor lighting quality and temporal instability. For LAV+TC, relit outputs degrade point cloud reconstruction, leading TC to produce artifacts from novel viewpoints. TL-Free suffers from a trade-off between fidelity and consistency. In contrast, our approach achieves a good balance of relighting quality, novel-view synthesis, and temporal stability, and outperforms all baselines in both fidelity and smoothness, as demonstrated in Table 1. Additional visual results are provided in Fig. Q in the Appendix. We further assess joint camera-illumination control using real in-the-wild videos as ground truth. As reported in Table 2, our method achieves the best PSNR, SSIM, LPIPS,

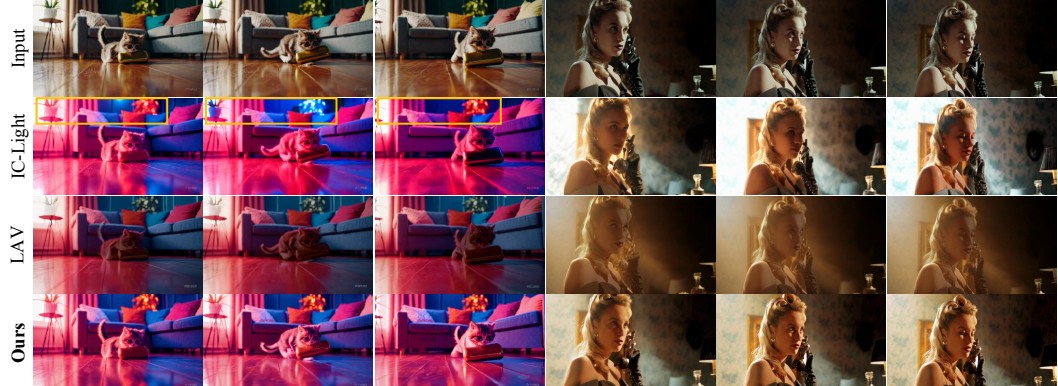

Figure 6: Qualitative comparison for video relighting with light prompts "neon light" (left) and "sunlight" (right). Our method outperforms baseline methods in both relighting quality and temporal consistency. Please refer to the supplementary video for clearer comparisons.

Table 4: Evaluation of video relighting using real in-the-wild videos as reference.

| Method | PSNR ↑ | SSIM ↑ | LPIPS ↓ | FVD ↓ |
|---|---|---|---|---|
| IC-Light | 11.75 | 0.517 | 0.422 | 67.50 |
| LAV | 12.66 | 0.530 | 0.429 | 74.64 |
| Ours | **13.84** | **0.581** | **0.369** | **56.60** |

Table 5: Quantitative results for background image-conditioned foreground video relighting. Methods marked with [*] are evaluated on the first 16 frames.

| Method | Image Quality | | Video Smoothness | | User Study (%, Ours) | | |
|---|---|---|---|---|---|---|---|
| | FID ↓ | Aesthetic ↑ | Motion Preservation ↓ | CLIP ↑ | RQ | VS | IP |
| IC-Light | / | 0.645 | 0.374 | 0.987 | 81.8 | 91.7 | 88.0 |
| Light-A-Video | 76.05 | 0.619 | 0.296 | 0.990 | 85.5 | 87.1 | 88.0 |
| Ours | **61.75** | **0.680** | **0.220** | **0.992** | / | / | / |
| RelightVid | 86.94 | 0.635 | 0.230 | 0.988 | 81.8 | 87.1 | 87.3 |
| Ours[*] | **56.60** | **0.682** | **0.199** | **0.990** | / | / | / |

and FVD scores, further confirming its advantages in both lighting fidelity and temporal consistency. User studies further validate these improvements.

### 4.3 VIDEO RELIGHTING RESULTS

**Text-conditioned relighting.** Fig. 6 and Fig. R in the Appendix show qualitative comparisons. Frame-wise IC-Light (Zhang et al., 2025b) achieves high single-frame quality but lacks temporal constraints, causing flicker in lighting and appearance. LAV (Zhou et al., 2025) integrates VDM (Wan et al., 2025) priors via a training-free fusion, improving stability but reducing fidelity. Our method attains both significant lighting accuracy and temporal coherence. Quantitative results in Table 3 confirm consistent gains in relighting fidelity and temporal smoothness, further supported by user studies. We additionally evaluate text-conditioned relighting using real in-the-wild videos as ground truth. As shown in Table 4, our method achieves the best performance across all metrics, further demonstrating its strengths in lighting fidelity and temporal stability.

**Background-conditioned relighting.** We also evaluate foreground video relighting conditioned on background images. As shown in Table 5, our method surpasses all baselines in both image quality and video smoothness. The corresponding qualitative analyses are provided in Sec. D.3. Additionally, we present results for HDR map-conditioned relighting (Sec. D.4) and reference image-conditioned relighting (Sec. D.5) in the Appendix.

Table 6: Qualitative ablation results for the joint camera-illumination control across different components: (a) training data, (b) architecture and lighting conditions, (c) training and conditioning strategy.

| Method | FID Score ↓ | Aesthetic ↑ | Motion Pres. ↓ | CLIP Score ↑ |
|---|---|---|---|---|
| (a.i) w/o static data | 123.35 | 0.594 | 3.749 | 0.987 |
| (a.ii) w/o dynamic data | 108.70 | 0.621 | 2.635 | 0.988 |
| (a.iii) w/o AI-gen data | 102.09 | 0.613 | 2.498 | 0.988 |
| (b.i) w/o fine-grained lighting cues | 143.02 | 0.602 | 2.242 | **0.989** |
| (b.ii) w/o global lighting control | 103.13 | 0.612 | 2.348 | **0.989** |
| (b.iii) light+text concat | 137.05 | 0.596 | 2.654 | **0.989** |
| (c.i) algorithm-generated GT | 137.83 | 0.524 | 4.066 | 0.986 |
| (c.ii) relit all frames | **71.10** | 0.571 | 4.238 | 0.986 |
| (c.iii) w/o soft mask | 148.51 | 0.545 | 2.879 | 0.988 |
| **Ours** | 101.06 | **0.623** | **2.007** | **0.989** |

## 4.4 ABLATION STUDIES

We conduct ablation studies on (a) training data, (b) architecture and lighting design, and (c) training and conditioning strategy. Quantitative results are in Table 6, and qualitative comparisons are shown in Fig. D of the Appendix. Removing static data (a.i) weakens unseen-view synthesis, as static videos provide natural cross-view pairs. Excluding dynamic data (a.ii) causes motion artifacts, while omitting AI-generated data (a.iii) lowers robustness to rare lighting, such as neon, where brightness may decay. Skipping fine-grained cues (b.i) limits the use of illumination priors from IC-Light (Zhang et al., 2025b), degrading relighting quality. Disabling global control (b.ii) causes fading or abrupt shifts under complex lighting, whereas adding it stabilizes results. Replacing our conditioning with light–text concatenation (b.iii), as in (Fang et al., 2025), also fails to leverage fine-grained lighting priors. Reversing supervision (c.i) by treating algorithm-generated outputs as ground truth harms fidelity, consistency, and novel-view synthesis. Relighting all frames instead of a single frame (c.ii) increases cost and reduces temporal coherence despite better FID. Discarding the soft mask (c.iii) blurs illumination domains and introduces interference, lowering overall performance.

## 5 CONCLUSION

We introduce Light-X, the first video generation framework that jointly controls camera trajectory and illumination from monocular videos. Our disentangled conditioning design leverages dynamic point clouds along user-defined trajectories to provide geometry and motion cues, while a relit frame is re-projected into the same geometry to provide illumination cues. To enable training, we further propose Light-Syn, a degradation-based data synthesis pipeline that constructs paired videos without requiring multi-view, multi-illumination captures. Extensive experiments show that Light-X consistently surpasses existing baselines in both joint camera–illumination control and video relighting, while flexibly adapting to diverse lighting conditions. We believe this work paves the way toward scalable generative modeling and controllable editing of complex real-world scenes.

**Limitations and Future Work.** 1) Our method relies on single-image relighting priors (*e.g.*, IC-Light (Zhang et al., 2025b)) to provide fine-grained lighting cues. In some scenes, the lighting quality of these priors may be suboptimal, which can in turn affect the quality of subsequent video generation. 2) The approach depends on point clouds as priors for novel camera viewpoints. When depth estimation is inaccurate, the resulting biased geometry may degrade generation quality, and the framework also struggles with very wide camera motions (*e.g.*, $360°$) due to limited 3D cues and the constrained generation length of the video diffusion model. 3) Like other video diffusion approaches, handling fine details (*e.g.*, hands) remains challenging, and the multi-step denoising process is computationally expensive. Future work may explore stronger video generation backbones (*e.g.*, Wan2.2 (Wan et al., 2025)) to enhance video quality, progressive point-cloud expansion to better support large camera ranges, and techniques such as Diffusion Forcing (Chen et al., 2024a) and Self Forcing (Huang et al., 2025) to extend video length.

**Acknowledgements.** This research is supported by cash and in-kind funding from NTU S-Lab and industry partner(s). This study is also supported by the Ministry of Education, Singapore, under its MOE AcRF Tier 2 (MOE-T2EP20221-0012, MOE-T2EP20223-0002).

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

## A  MORE IMPLEMENTATION DETAILS

### A.1  TRAINING-FREE BASELINE: TL-FREE

The baseline method TL-Free is inspired by Light-A-Video (LAV) (Zhou et al., 2025), which integrates TrajectoryCrafter (YU et al., 2025) and IC-Light (Zhang et al., 2025b) in a training-free manner. LAV introduces three key components: consistent light attention, progressive light fusion, and details compensation, to achieve temporally coherent video relighting. However, unlike LAV, which directly processes original video frames for relighting, our objective is to simultaneously perform relighting and camera viewpoint changes. Thus, the model input is not the raw video but point-cloud projected views, which are geometrically aligned yet inevitably contain black borders and holes. This critical difference necessitates adapting the LAV modules as follows: **1) Details compensation.** While effective in LAV for enhancing frame-level fidelity, this module relies on the input video to supplement missing details. For our projected views, however, it propagates black borders and holes, severely degrading the results. We thus discard this module in TL-Free. **2) Progressive light fusion.** In LAV, the fusion ratio between IC-Light and the video diffusion backbone is controlled by a schedule: IC-Light dominates early denoising steps and gradually diminishes. For our projected views, applying IC-Light early is problematic, since relighting images with black borders or incomplete regions is ill-posed. We therefore disable IC-Light fusion during early denoising and only activate it in later steps, once the model has filled in missing content. **3) Consistent light attention.** This module is retained, as it ensures temporally consistent illumination across frames and remains effective even when operating on projected views.

### A.2  BACKGROUND-CONDITIONED CONTROL

As shown in Fig. A, our framework supports background-conditioned video relighting. Specifically, the source video $V^s$ is obtained by fusing a foreground video with a background video using foreground masks. IC-Light (Zhang et al., 2025b) is then applied to generate a sparse relit video $\hat{V}^s$, which provides illumination cues. Finally, both $V^s$ and $\hat{V}^s$ are fed into our model to produce the relit video with consistent illumination and motion.

### A.3  BASELINE

For Light-A-Video (LAV) (Zhou et al., 2025), we adopt the officially released Wan2.1 (Wan et al., 2025) implementation and use the default hyperparameter settings. For AnyV2V (Ku et al., 2024), we evaluate the officially released model built on I2VGen-XL (Zhang et al., 2023) under its default configuration. The released model generates 16-frame videos at a resolution of $512 \times 512$. For RelightVid (Fang et al., 2025), as it currently only provides a background-image-conditioned model without text-conditioned variants or training code, we evaluate it exclusively under the background-conditioned setting using the official model released by the authors. Its outputs also have 16 frames with a spatial resolution of $512 \times 512$. For fair comparison, all baseline outputs are uniformly resized to match the resolution adopted in our evaluation.

### A.4  EVALUATION PROTOCOL AND USER STUDY

**Evaluation Protocol.** For text-conditioned relighting, we randomly select one lighting prompt (*e.g.*, sunlight, soft light, neon light, or red and blue neon light) and one lighting direction (top, bottom, left, or right) for each video. For novel-view video generation, one of four predefined camera trajectories is randomly chosen. After the condition is determined, we apply the same lighting prompt and camera trajectory to all methods to ensure fair comparison. In the joint camera–illumination control evaluation, we first employ TrajectoryCrafter (YU et al., 2025) to generate the novel-view sequence. This sequence is then compared against relit sequences produced by other methods under the same trajectory to compute the flow error (*i.e.*, Motion Preservation). Additionally, we apply IC-Light (Zhang et al., 2025b) to relight the novel-view video from TrajectoryCrafter, which serves as the reference for calculating FID with respect to the relit outputs of all baselines.

**User Study.** We conducted a user study to evaluate the effectiveness of our method across three tasks: 1) joint camera–illumination control, 2) text-conditioned video relighting, and 3) background-conditioned video relighting. The study was conducted online, and screenshots of the interface

are shown in Fig. B. The interface displayed the input video, the corresponding relighting prompt (text or background image), and two relit results (denoted as Method 1 and Method 2) side by side. Participants could play both videos in parallel and directly compare their quality. On the left panel, four criteria were listed with radio buttons for selection: Relighting Quality (RQ, lighting fidelity, and alignment with the condition), Video Smoothness (VS, temporal stability across frames), Identity Preservation (IP, consistency of the object's identity and appearance), and 4D Consistency (4DC, spatio-temporal coherence under novel-view settings). For each criterion, participants were required to select which method performed better. They were also allowed to choose "Hard to judge" or skip to the next example if necessary. To reduce fatigue and ensure reliable feedback, the system required participants to submit responses after completing 10 groups of comparisons. The study was conducted anonymously, and no personally identifiable data were collected. In total, we collected responses from 57 participants.

## B    DETAILED DATA CURATION

**Training Data Requirements Analysis.** As discussed in Sec. 3.3 in the main text, training our framework requires an input video $V^s$, a paired target video $V^t$, and conditioning sequences $V^p$, $\hat{V}^s$, and $\hat{V}^p$. To ensure effective training, these modalities should satisfy the following requirements:

- **Target video $V^t$.** Serving as the ground truth, the target video should be of high visual quality and exhibit temporal consistency.

- **Input video $V^s$.** Serving as the reference sequence injected into the network, the input video should remain 4D-consistent with the target video $V^t$ in their overlapping regions.

- **Projected source views $V^p$.** Serving as a geometric view-transformation prior, these projections should maintain content consistency with the target video $V^t$ in shared visible regions.

- **Sparse relit video $\hat{V}^s$.** Serving as an explicit lighting prior for the diffusion model, the relit frame should share the same illumination as the target video $V^t$.

- **Projected relit views $\hat{V}^p$.** Serving as fine-grained illumination cues, these projections should be geometrically aligned with the corresponding projected source views $V^p$, ensuring that illumination information is accurately fused with the geometric prior.

**Pipeline Design.** As shown in Fig. C, we design a degradation-based pipeline to construct paired videos based on these requirements. Specifically, we treat an in-the-wild video as the target sequence $V^t$ and generate its degraded counterpart as the input sequence $V^s$ to satisfy the above constraints. Furthermore, by recording the transformations applied during the degradation process, we apply their inverses to map the geometry and illumination of the target video back to the degraded sequence, thereby producing the corresponding conditioning cues $V^p$, $\hat{V}^s$ and $\hat{V}^p$ that conform to the training requirements and enable training with the degraded video as input.

**Data Sources.** We curate training pairs from three complementary sources:

- **Static scenes.** Monocular videos of static scenes naturally provide multi-view observations of the same scene. We adopt two strategies to construct paired samples with only varied viewpoints: 1) sample a video clip as one view and create the other by repeating a randomly selected frame from the same sequence; 2) select two clips with overlapping content as a pair. For both cases, we employ VGGT (Wang et al., 2025b) to reconstruct depths and camera poses, thereby establishing the geometric transformations between views. To further introduce illumination variation, we process the data according to the pairing type: 1) relight the image-repeated video using IC-Light (Zhang et al., 2025b), which serves as the degraded input $V^s$ while naturally maintaining temporal consistency due to the repeated frames, and take the other video as $V^t$; 2) relight one clip with Light-A-Video (LAV) (Zhou et al., 2025), treating the relit clip as $V^s$ and the remaining clip as $V^t$. Although the latter approach yields slightly weaker temporal consistency, it still preserves the scene content and meets our data requirements. Finally, leveraging the geometric transformations estimated by (Wang et al., 2025b), we warp the information in $V^s$ to the viewpoint of $V^t$, thereby constructing the data required for training. With this approach, we curate 8k static training samples from the DL3DV (Ling et al., 2024) dataset.

- **Dynamic scenes.** Given a dynamic monocular video $\boldsymbol{V}^t$, we construct degraded counterparts $\boldsymbol{V}^s$ using three strategies: 1) relight $\boldsymbol{V}^t$ with Light-A-Video (Zhou et al., 2025) and then synthesize a novel-view video $\boldsymbol{V}^s$ via TrajectoryCrafter (YU et al., 2025); 2) synthesize a novel-view video using (YU et al., 2025) and then apply (Zhou et al., 2025) to introduce illumination variations, producing $\boldsymbol{V}^s$. 3) directly generate a relit and novel-view video $\boldsymbol{V}^s$ through our designed training-free pipeline TL-Free (details are provided in the Sec. A.1). The degraded videos $\boldsymbol{V}^s$ produced by these strategies are used as inputs for model training. Although their temporal consistency and visual quality are not perfect, they satisfy our data requirements, such as maintaining content consistency with the target video $\boldsymbol{V}^t$ in overlapping regions. To ensure geometric alignment, all warping operations rely on depths estimated once from the original video rather than being re-estimated from intermediate results. During degradation, we derive the correspondence flow $F_{t \to s}$ from the depth and relative pose and warp the original frame to the degraded view. When constructing the training set, we then apply the reverse flow $F_{s \to t}$ to warp the degraded samples back to the original viewpoint, thereby obtaining geometrically aligned conditions. Using this procedure, we curate 8k dynamic training samples from the VDW (Wang et al., 2023) dataset.

- **AI-generated videos.** While the above methods use high-quality real-world videos as supervision, most videos exhibit relatively uniform and soft lighting, limiting the diversity of illumination conditions. To address this, we design a data pipeline based on commercial video generation models to synthesize videos with richer lighting variations. Specifically, we first employ (YU et al., 2025) to generate a novel-view video from the original sequence $\boldsymbol{V}^t$, then extract its first frame and relight it using (Zhang et al., 2025b). The relit frame, together with the novel-view video, is fed into the first-frame-guided video-to-video mode of a commercial generative model (*e.g.*, Runway or Luma) to produce a relit video, resulting in the paired video $\boldsymbol{V}^s$. This approach yields videos with diverse illumination while maintaining high temporal consistency, thanks to the commercial model's powerful capability. For this set of data, we follow the standard training strategy, using $\boldsymbol{V}^t$ as the input and $\boldsymbol{V}^s$ as the target video. However, a limitation of this approach is that, although it ensures temporal consistency, the commercial model tends to generate content inconsistent with the original video when the scene or camera motion is relatively large, violating the data requirements discussed above and adversely affecting the learning of our model. Therefore, we only retain videos with small motion, which we identify and filter using an optical-flow-based criterion (Huang et al., 2024), and we ultimately curate 2k samples from the OpenVid-1M (Nan et al., 2024) dataset.

Together, the three data sources provide complementary training pairs for our framework: static scenes offer accurate multi-view data, dynamic scenes supply samples with scene motion, and AI-generated videos enrich illumination diversity. All of them satisfy the training requirements, providing paired inputs, targets, and geometrically aligned conditioning cues.

## C  PRELIMINARY: VIDEO DIFFUSION MODELS

Video diffusion models consist of two stages: a forward process and a reverse process. The forward process starts from clean video data $\boldsymbol{x}_0 \in \mathbb{R}^{f \times 3 \times h \times w}$ and gradually injects noise to create noisy states as $\boldsymbol{x}_t = \alpha_t \boldsymbol{x}_0 + \sigma_t \epsilon$, where $\epsilon \sim \mathcal{N}(\mathbf{0}, \mathbf{I})$ and $\alpha_t^2 + \sigma_t^2 = 1$. The reverse process removes noise with a predictor $\epsilon_\theta(\boldsymbol{x}_t, t)$, optimized by

$$\min_\theta \mathbb{E}_{t \sim \mathcal{U}(0,1), \, \epsilon \sim \mathcal{N}(\mathbf{0}, \mathbf{I})} \left[ \|\epsilon_\theta(\boldsymbol{x}_t, t) - \epsilon\|_2^2 \right]. \tag{8}$$

For computational efficiency, videos are first compressed into latents $\boldsymbol{z} = \mathcal{E}(\boldsymbol{x})$ using a pre-trained 3D VAE (Kingma & Welling, 2013; Rombach et al., 2022). The latents are then patchified, concatenated with text embeddings, and fed into the noise estimator. Recent works (Yang et al., 2024b; Wan et al., 2025; Lin et al., 2024; Kong et al., 2024; Brooks et al., 2024; Team, 2024) commonly adopt the Diffusion Transformer (DiT) (Peebles & Xie, 2023) as the noise estimator, owing to its strong modeling capacity and flexible scalability. During inference, noisy latents are iteratively denoised, then reconstructed by the VAE decoder to produce the final video $\hat{\boldsymbol{x}} = \mathcal{D}(\boldsymbol{z})$.

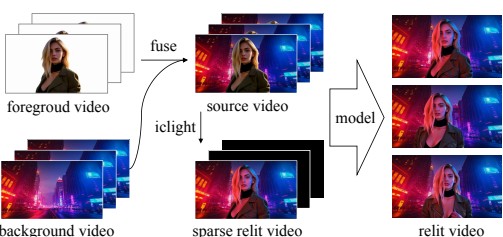

Figure A: Overview of background-conditioned video relighting. A foreground video is fused with a background video to form the source video, while IC-Light generates a sparse relit video. Both are fed into our model to produce the final relit video with consistent illumination and motion.

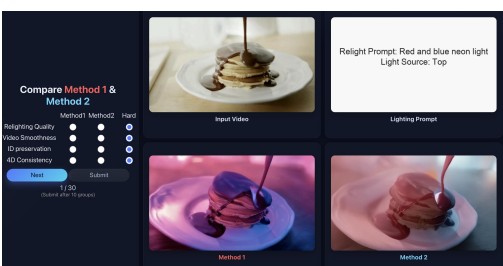

Figure B: The web interface of our user studies. Participants were shown the input video, the relighting prompt (text or background image), and results of two methods (Method 1 and Method 2) side by side. They evaluated each pair across four criteria by selecting the better method.

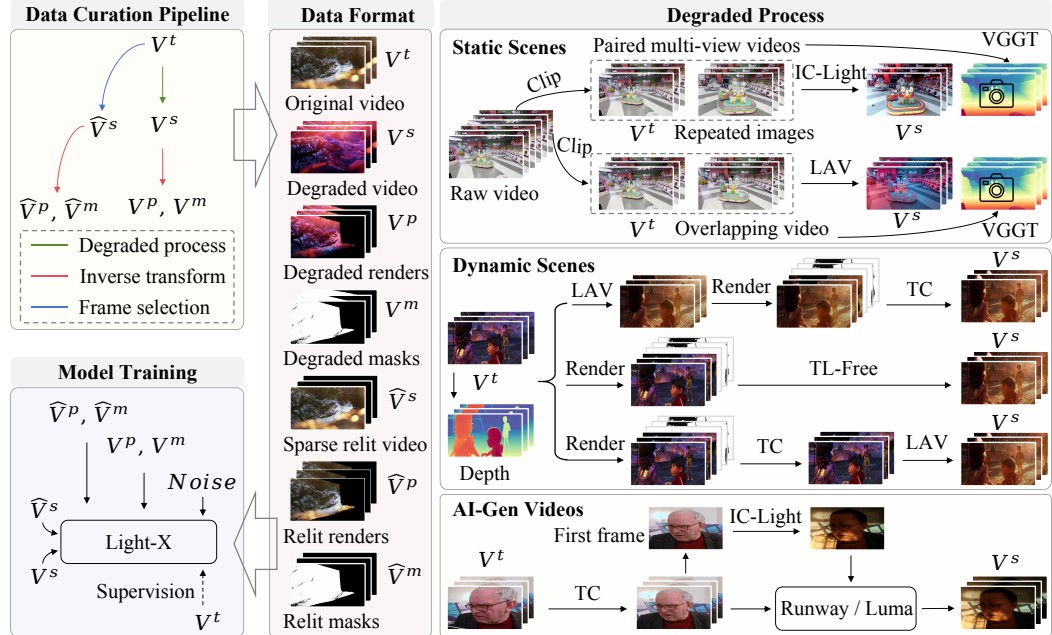

Figure C: Detailed Data curation pipeline **Light-Syn**. Given an original in-the-wild video $V^t$, we synthesize a degraded counterpart $V^s$ using different strategies for static, dynamic, and AI-generated scenes. From $V^s$, we obtain geometry-aligned renders and masks $(V^p, V^m)$ and relit counterparts $(\hat{V}^p, \hat{V}^m)$ via inverse geometric transformations. The curated data provide paired videos for training, with Light-X taking the degraded video as input, the other signals as conditions, and the original video as ground truth.

# D  MORE EXPERIMENTAL RESULTS

## D.1  ADDITIONAL RESULTS ON CAMERA–ILLUMINATION CONTROL

**Additional Visual Comparisons.** Fig. Q presents further results on joint camera–illumination control. TC (YU et al., 2025)+LAV (Zhou et al., 2025) inherits the limited relighting capacity of LAV, which becomes particularly fragile under large camera motion, leading to distorted illumination and unstable temporal transitions. When LAV is applied after TC, the relit results undermine the geometric reconstruction, causing TC to introduce noticeable artifacts in novel views. TL-Free, on the other hand, fails to balance fidelity and consistency, often yielding either excessively simplified

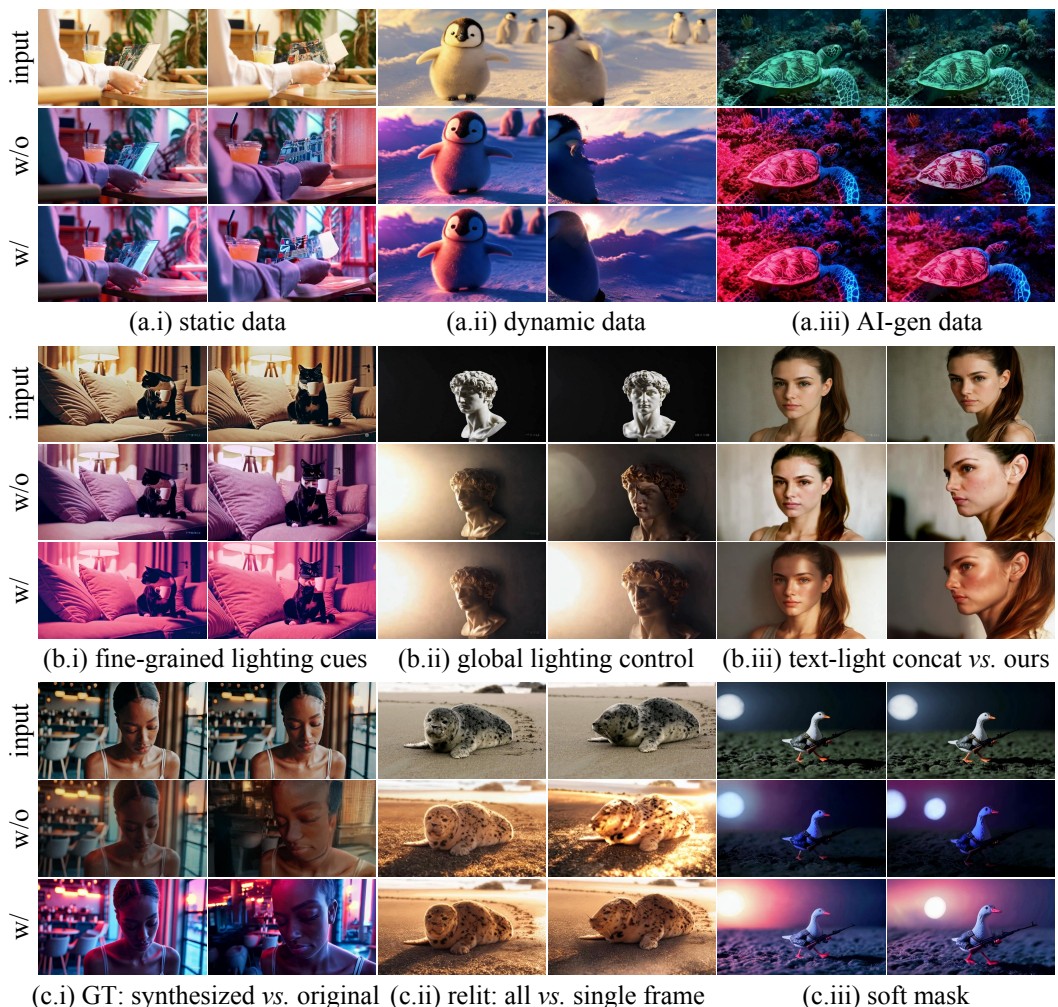

Figure D: Qualitative ablation results for the joint camera-illumination control task.

Table A: Quantitative comparison with baseline methods for joint camera-illumination control on the iPhone (Gao et al., 2022) multi-view dataset.

| Method | FID ↓ | Aesthetic ↑ | Motion Preservation ↓ | CLIP ↑ |
|---|---|---|---|---|
| Target-view video + IC-Light | / | **0.571** | 5.412 | 0.973 |
| TrajectoryCrafter + Light-A-Video | 236.57 | 0.516 | 5.757 | 0.987 |
| Light-A-Video + TrajectoryCrafter | 218.72 | 0.520 | 4.620 | 0.986 |
| TL-Free | 198.55 | 0.488 | 6.428 | 0.985 |
| Ours | **155.36** | 0.557 | **3.316** | **0.987** |

Table B: Quantitative comparison of video relighting under HDR-map conditioning.

| Method | PSNR ↑ | SSIM ↑ | LPIPS ↓ | Consistency (Temp. / Subj. / Backg.) ↑ | Motion Smooth. ↑ | Aesthetics (Qual. / Img.) ↑ |
|---|---|---|---|---|---|---|
| DiffusionRenderer | 11.88 | 0.4510 | 0.4931 | 0.9921 / 0.9560 / 0.9599 | 0.9944 | 0.5262 / 0.4495 |
| Ours | 16.98 | 0.6653 | 0.2504 | 0.9933 / 0.9608 / 0.9676 | 0.9927 | 0.5909 / 0.6277 |

lighting effects or severe temporal flickering. In contrast, our method achieves high-quality relighting

under diverse lighting prompts (*e.g.*, neon, soft, sunlight) while preserving temporal consistency and realistic novel-view content generation, consistently outperforming baseline methods.

**Results on iPhone Multi-view Dataset.** We further adopt the iPhone dataset (Gao et al., 2022), which contains 7 dynamic scenes captured with a casually moving camera and two static cameras. Following prior work (YU et al., 2025; Wang et al., 2024a), we discard the "Space-out" and "Wheel" scenes due to camera and LiDAR errors, and use the remaining 5 refined scenes, namely *Apple*, *Block*, *Paper*, *Spin*, and *Teddy*. These data provide multi-view videos of the same dynamic scenes, serving as a valuable benchmark for assessing novel-view content generation with or without relighting. In our evaluation, the casually moving camera videos are used as input to synthesize target static-camera views with relighting, while the ground-truth static-camera videos serve as references for computing motion preservation. For FID computation, we relight the ground-truth static-camera videos using IC-Light (Zhang et al., 2025b) as the reference. As shown in Table A, our method achieves the best overall performance on this dataset. In particular, it attains the lowest FID, indicating superior visual fidelity, and significantly improves motion preservation compared to baselines. Although IC-Light applied to target-view videos yields a slightly higher aesthetic score, since it does not require novel-view generation, our approach achieves a better overall balance across all metrics, consistently outperforming baselines in relighting quality, temporal stability, and novel-view generation.

## D.2 Additional Results on Text-Conditioned Relighting

We provide more qualitative results of text-conditioned video relighting in Fig. R. Frame-wise IC-Light (Zhang et al., 2025b) delivers high-quality relighting on individual frames, but the absence of temporal modeling leads to noticeable flickering across videos. LAV (Zhou et al., 2025) leverages video diffusion priors through a training-free fusion strategy, which improves temporal stability but often compromises lighting fidelity and detail. In contrast, our method achieves high-quality and consistent vdieo relighting under diverse lighting prompts while preserving temporal consistency, consistently outperforming baseline methods.

## D.3 Additional Results on Background-Conditioned Relighting

We further present qualitative comparisons of background image-conditioned video relighting in Fig. E. IC-Light (Zhang et al., 2025b) often produces inconsistent illumination across frames due to its frame-wise nature, resulting in flickering and mismatched tones. RelightVid (Fang et al., 2025) improves temporal stability but tends to over-smooth the lighting effects, leading to a loss of realism. LAV (Zhou et al., 2025) enhances consistency but sacrifices fine details, producing less faithful relighting results. In contrast, our method effectively integrates the subject with the target background illumination, achieving natural lighting and temporally stable outputs. As shown in Fig. F, our framework generalizes well to diverse background images, adapting the relighting smoothly while preserving subject identity and fine-grained details.

## D.4 Additional Results on HDR Map-Conditioned Relighting

We further present qualitative results of HDR map-conditioned video relighting in Fig. S. Given an input video and an HDR environment map, our model generates plausible relit videos that faithfully reflect the target illumination, demonstrating the potential of our soft-mask design to adapt to diverse lighting conditions. To quantitatively evaluate this setting, we also compare against DiffusionRenderer (Liang et al., 2025). Since no paired in-the-wild dataset is available, we adopt a strategy similar to our data curation pipeline. Specifically, for each in-the-wild video in the evaluation set, we extract HDR maps using DiffusionLight (Phongthawee et al., 2024), and use Light-a-Video (Zhou et al., 2025) to produce aligned relit videos. These relit videos are then used as model inputs, while the extracted HDR maps serve as conditions. We compute metrics (*e.g.*, PSNR, SSIM, LPIPS) between the model outputs and the original in-the-wild videos, where our approach achieves superior scores. However, this evaluation setting inherently favors our training paradigm. To provide a more balanced assessment, we further evaluate video quality on VBench (Huang et al., 2024). The results show that our outputs are comparable to DiffusionRenderer in terms of overall visual quality, demonstrating the generalization ability of our framework to diverse lighting conditions and the potential of HDR maps as a versatile control signal.

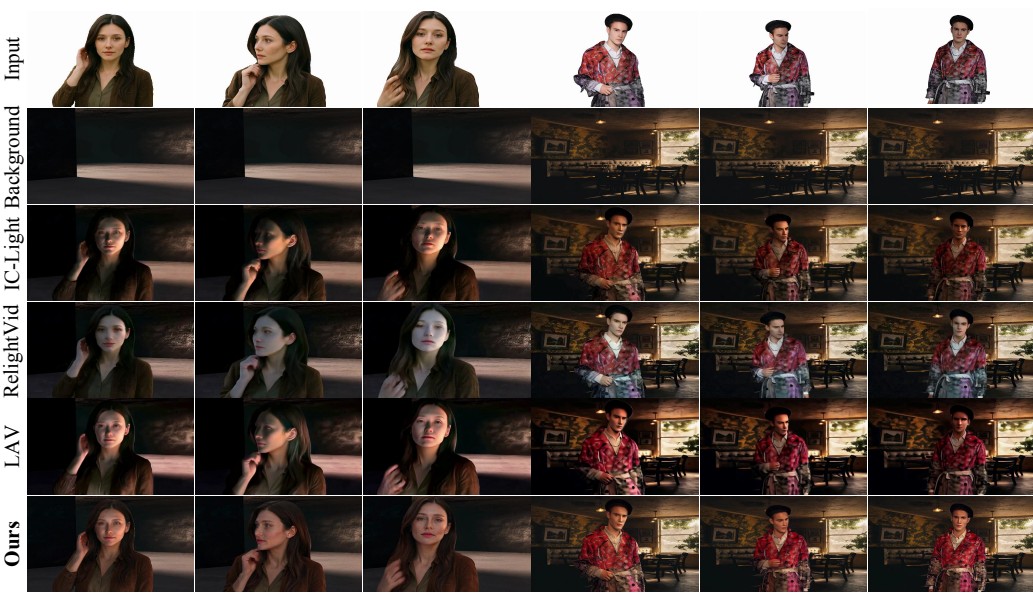

Figure E: Qualitative comparison of background image-conditioned video relighting. Our method achieves superior both relighting quality and temporal consistency compared to baseline methods.

### D.5    ADDITIONAL RESULTS ON REFERENCE IMAGE-CONDITIONED RELIGHTING

We present the results of reference image-conditioned video relighting in Fig. T, where a single reference image specifies the target illumination style to be transferred to the input video. To the best of our knowledge, our framework is the first to enable this setting. Furthermore, as shown in Fig. U, our approach also supports simultaneous relighting and novel-view synthesis, achieving both illumination control and camera trajectory manipulation.

### D.6    ADDITIONAL RESULTS ON NOVEL VIEW SYNTHESIS

To further evaluate novel-view video generation, we compare our method with TrajectoryCrafter (YU et al., 2025) on the multi-view iPhone (Gao et al., 2022) dataset. As shown in Table C, our method achieves higher PSNR and lower LPIPS, while maintaining comparable SSIM. This suggests that our framework performs on par with its baseline method in novel-view video synthesis, while providing additional flexibility for joint camera–illumination control.

Table C: Quantitative comparison between our method and TrajectoryCrafter (YU et al., 2025) on the iPhone (Gao et al., 2022) dataset.

| Method | PSNR ↑ | SSIM ↑ | LPIPS ↓ |
|---|---|---|---|
| TrajectoryCrafter | 14.6204 | 0.5725 | 0.3801 |
| Ours | 15.6016 | 0.5696 | 0.3519 |

### D.7    GEOMETRY CONSISTENCY EVALUATION

To further validate the geometric coherence of our relighting results, we conduct a comprehensive point-cloud-based evaluation. Specifically, we reconstruct dynamic and static 3D geometry from both the input and relighted videos using two state-of-the-art methods: the dynamic reconstruction method MegaSAM (Li et al., 2025b) and the static reconstruction model VGGT (Wang et al., 2025b). For each video, we extract per-frame point clouds and compute the Chamfer Distance (CD) between the input and relighted reconstructions. We report multiple statistics, including *Mean*, *Median*, *Standard Deviation*, *Minimum*, and *Maximum* Chamfer Distance (CD), to provide a comprehensive assessment of geometric discrepancies. As shown in Table D, our method achieves the lowest mean and median CD, demonstrating substantially better preservation of geometric structure. Qualitative point cloud visualizations are included in the supplementary video.

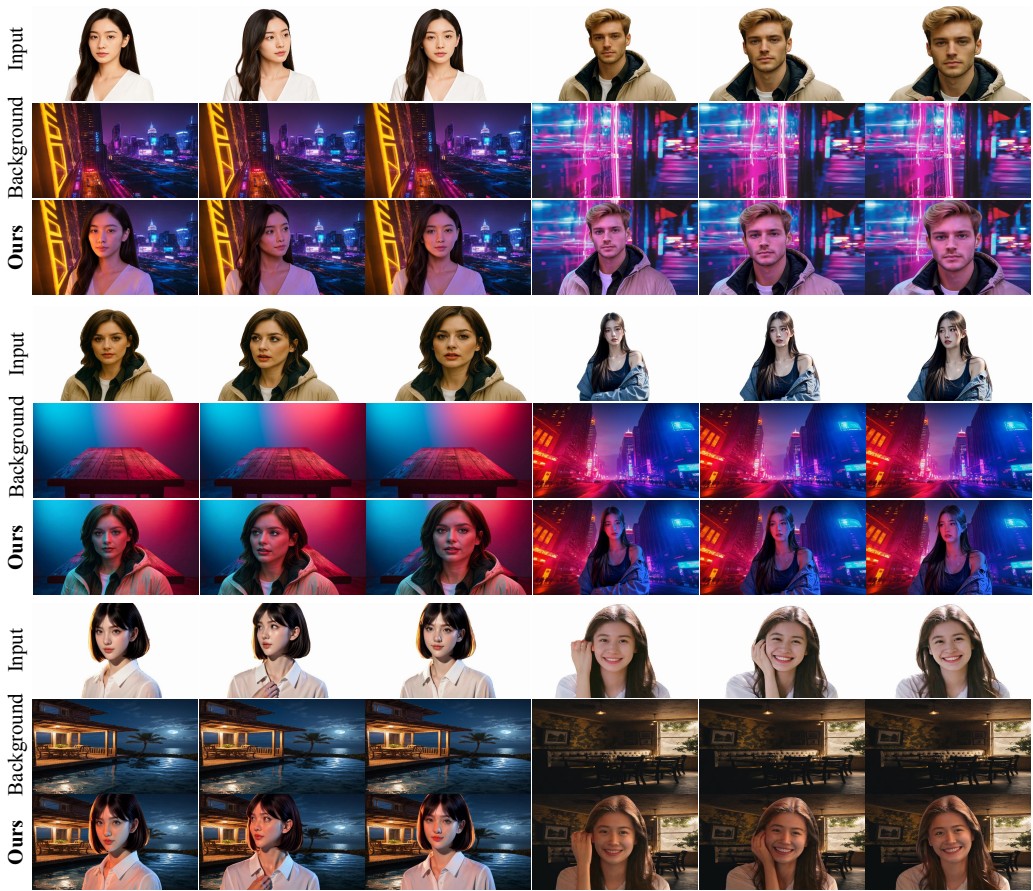

Figure F: Visual results of background image-conditioned video relighting. Our method adapts the foreground subject to diverse background images, producing natural illumination and consistent appearance across frames.

Table D: Geometry consistency evaluation. Chamfer Distance (CD) between point clouds reconstructed from the input and relighted videos. * denotes evaluation on the first 16 frames.

| Method | Avg CD ↓ | Median ↓ | Std ↓ | Min ↓ | Max ↓ |
|---|---|---|---|---|---|
| IC-Light | 0.5012 | 0.1933 | 1.3300 | 0.0056 | 15.6113 |
| LAV | 0.8979 | 0.1903 | 4.5258 | 0.0096 | 58.0839 |
| Ours | **0.3753** | **0.1581** | 0.7228 | 0.0063 | 5.9780 |
| IC-Light + AnyV2V | 0.8896 | 0.2813 | 1.8282 | 0.0055 | 19.9080 |
| Ours* | **0.3784** | **0.1577** | 0.7535 | 0.0064 | 5.8883 |

## D.8 ANALYSIS ON NON-LAMBERTIAN SURFACES

As shown in Fig. G, we present additional results on non-Lambertian surfaces, showing that the model preserves specular effects without washing out fine details. The corresponding video results are provided in the supplementary video.

## D.9 LARGE CAMERA TRAJECTORIES

Similar to other approaches (Yu et al., 2024; YU et al., 2025; Liu et al., 2025a) that condition on point-cloud priors, our method relies on point-cloud cues to guide viewpoint-consistent generation. Under extremely wide camera motions, these cues become sparse or even unavailable, which can

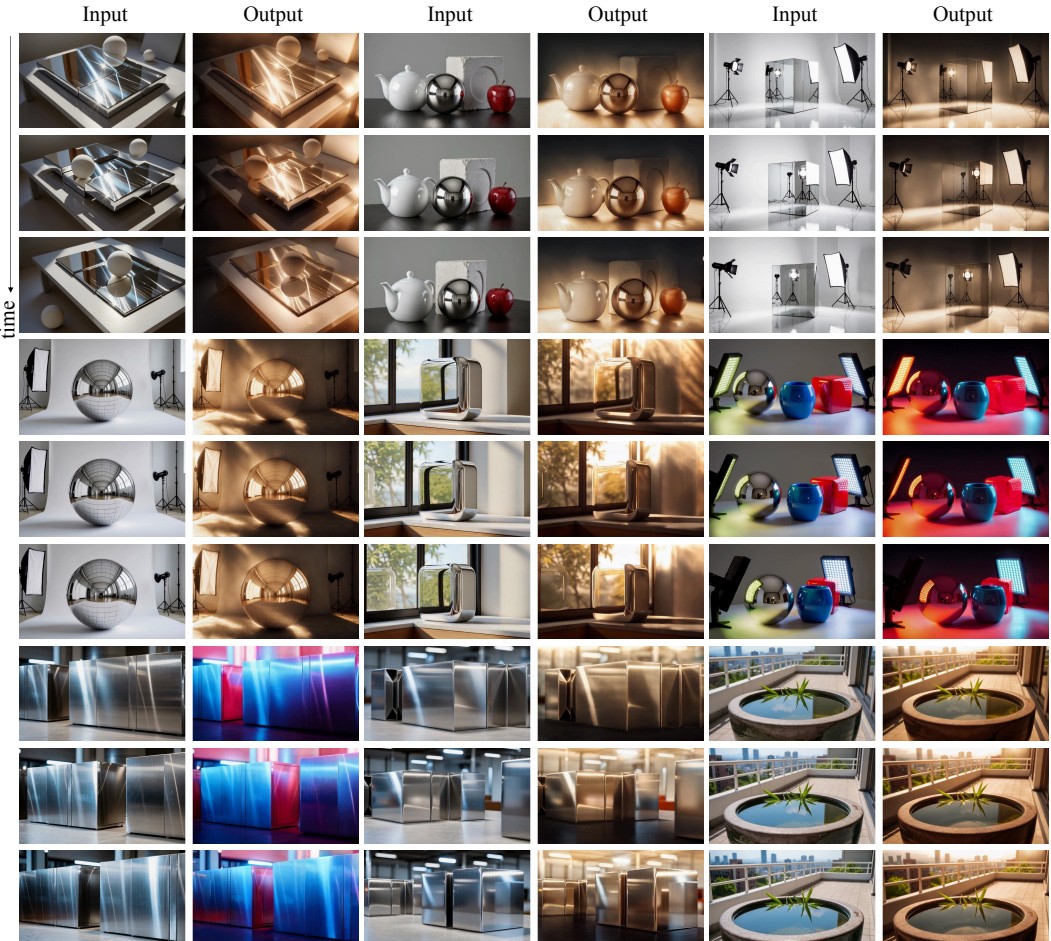

Figure G: Results on non-Lambertian surfaces.

negatively affect synthesis quality. Nevertheless, the model remains robust under substantial viewpoint deviations, roughly up to $60°$, as shown in Fig. H. The corresponding video results are provided in the supplementary video.

### D.10 COMPARISON WITH MORE RECENT METHODS

We further compare Light-X with several recent systems that provide either camera control or lighting control capabilities, covering the latest advances in controllable video generation. For camera control, we include ReCamMaster (Bai et al., 2025), which does not rely on explicit 3D representations, and Free4D (Liu et al., 2025a), which incorporates an explicit 3D representation. For lighting control, we further compare against TC-Light (Liu et al., 2025b). The results for joint camera–illumination control and video relighting are reported in Tables E and F. Across all comparisons, Light-X maintains superior image fidelity, aesthetic quality, and motion consistency, demonstrating strong state-of-the-art performance even against these recent systems. Since Free4D requires per-scene optimization, typically taking over an hour per scene, we evaluate it on the 10 scenes provided on its official project page for practical comparison. The corresponding results are shown in Table G.

### D.11 FID DEGRADATION WITH TEMPORAL DISTANCE

We analyze how FID changes with increasing temporal distance from the relit reference frame. As shown in Fig. J, FID gradually increases because it is computed against the IC-Light (Zhang et al.,

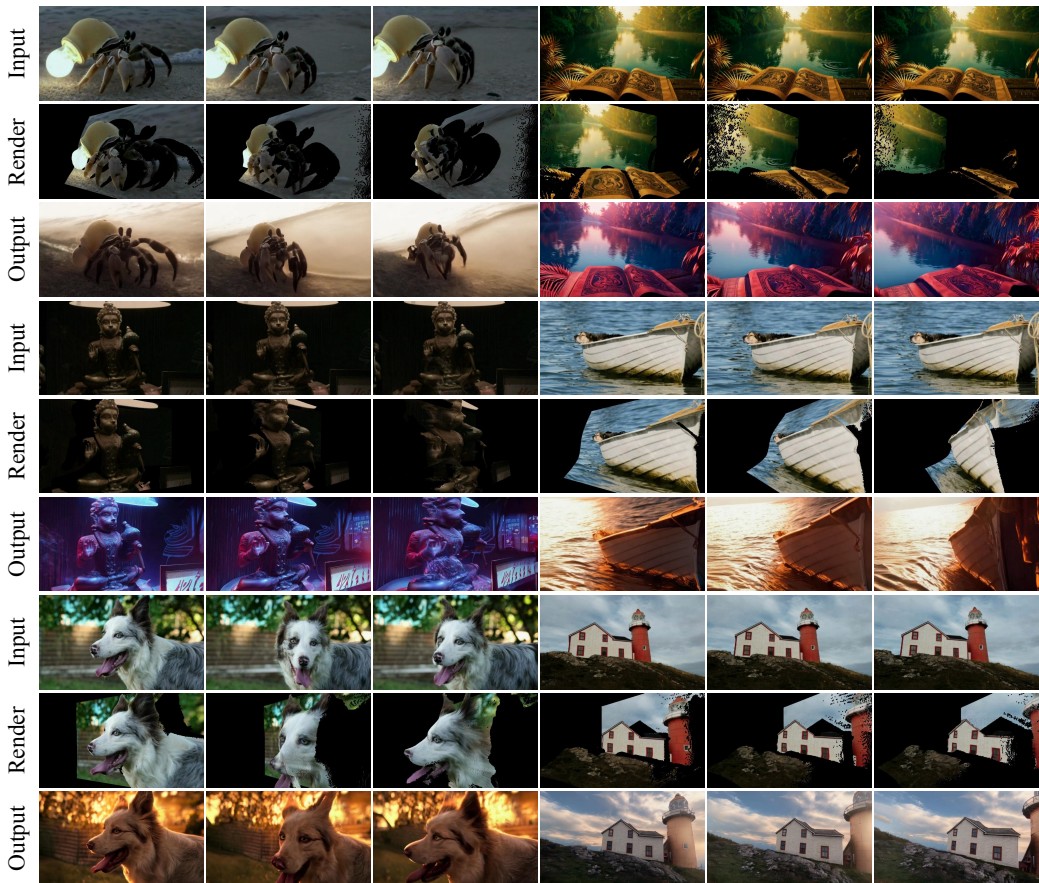

Figure H: Qualitative results under large camera trajectories.

Table E: Comparison with more recent methods for joint camera-illumination control.

| Method | FID ↓ | Aesthetic ↑ | Motion Pres. ↓ | CLIP ↑ |
|---|---|---|---|---|
| TC + IC-Light | / | 0.573 | 6.558 | 0.976 |
| TC + LAV | 138.89 | 0.574 | 4.327 | 0.986 |
| LAV + TC | 144.61 | 0.596 | 5.027 | 0.987 |
| TL-Free | 122.73 | 0.595 | 3.356 | 0.987 |
| ReCam + IC-Light | / | 0.513 | 6.511 | 0.973 |
| LAV + ReCam | 163.56 | 0.514 | 7.259 | **0.989** |
| ReCam + LAV | 152.03 | 0.501 | 3.157 | 0.987 |
| TC + TC-Light | 154.99 | 0.534 | 4.276 | 0.986 |
| TC-Light + TC | 161.76 | 0.555 | 5.563 | 0.988 |
| Ours | **101.06** | **0.623** | **2.007** | **0.989** |

2025b) relit reference: the first frame is directly relit by IC-Light, whereas later frames rely on Light-X to propagate illumination cues over time.

For video relighting, FID increases smoothly from 38 to 82 over 49 frames. For joint camera-illumination control, FID rises from 56 to 100. Importantly, even the last frame still outperforms baseline methods such as LAV (Zhou et al., 2025) (FID: 112.45) and TL-Free (FID: 122.73).

Table F: Comparison with more recent lighting control methods on video relighting.

| Method | FID ↓ | Aesthetic ↑ | Motion Pres. ↓ | CLIP ↑ |
|---|---|---|---|---|
| IC-Light | / | 0.632 | 3.293 | 0.983 |
| LAV | 112.45 | 0.614 | 2.115 | 0.991 |
| TC-Light | 144.32 | 0.546 | 1.657 | 0.991 |
| Ours | **83.65** | **0.645** | **1.137** | **0.993** |

Table G: Evaluation results on the 10 scenes released on the Free4D (Liu et al., 2025a) project page.

| Method | FID ↓ | Aesthetic ↑ | Motion Pres. ↓ | CLIP ↑ |
|---|---|---|---|---|
| Free4D + IC-Light | / | 0.576 | 0.823 | 0.990 |
| Free4D + LAV | 98.85 | 0.574 | 0.549 | 0.996 |
| Ours | **73.98** | **0.583** | **0.349** | **0.997** |

### D.12 Performance with Occluded Relit Frames

As shown in Fig. I, Light-X remains robust even when the relit reference frame (the first frame in the illustrated example) is partially occluded or contains incomplete scene information. Illumination cues are propagated coherently even under occlusions such as a book or a mask covering parts of the face. Furthermore, in zoom-out scenarios where later frames reveal previously unseen regions, Light-X continues to produce reasonable and consistent relighting for these newly visible areas. The corresponding video results are provided in the supplementary video.

### D.13 Robustness to Depth Noise

Since Light-X relies on projected point-cloud views as soft geometric cues, inaccuracies in depth estimation may introduce biased geometry and affect performance. Nevertheless, the method does not require highly accurate depth and remains robust under moderate noise levels. We conducted a controlled experiment on 12 randomly selected scenes. Using DepthCrafter (Hu et al., 2024) as the default depth estimator, we injected Gaussian noise into the depth maps:

$$\tilde{D} = D + \epsilon \cdot D, \quad \epsilon \sim \mathcal{N}(0, \text{ rate}),$$

where *rate* controls the perturbation strength. The performance under different noise levels is summarized in Table H. As shown in Fig. K, Light-X maintains coherent illumination and motion consistency even when depth maps are perturbed with moderate Gaussian noise. Light-X degrades gracefully as noise increases and consistently outperforms baseline methods. Corresponding qualitative video results are provided in the supplementary video.

### D.14 Choice of the Relighting Reference Frame

Light-X does not rely on using the first frame as the relighting reference. During training, the relit frame is randomly sampled, and during inference any frame may be selected. In the main text, we adopt the first frame purely for implementation convenience and reproducibility.

To assess robustness to the choice of reference frame, we evaluate four strategies ("first", "middle", "last", "random") on 200 videos. As shown in Table I, Light-X achieves similar performance across all strategies, showing that the method is robust to the selection of the relighting reference frame.

### D.15 Using Light-X for Data Generation

We investigate whether Light-X can be used to generate higher quality relit videos to further improve the training data. To this end, we relight 2k samples from the training set using Light-X and fine-tune the original model with this additional data. As shown in Table J, the fine-tuned model shows only marginal improvements, suggesting that the original Light-Syn data already provides sufficiently strong supervision and that real in-the-wild videos remain the primary source of learning signals.

Input      Output      Input      Output

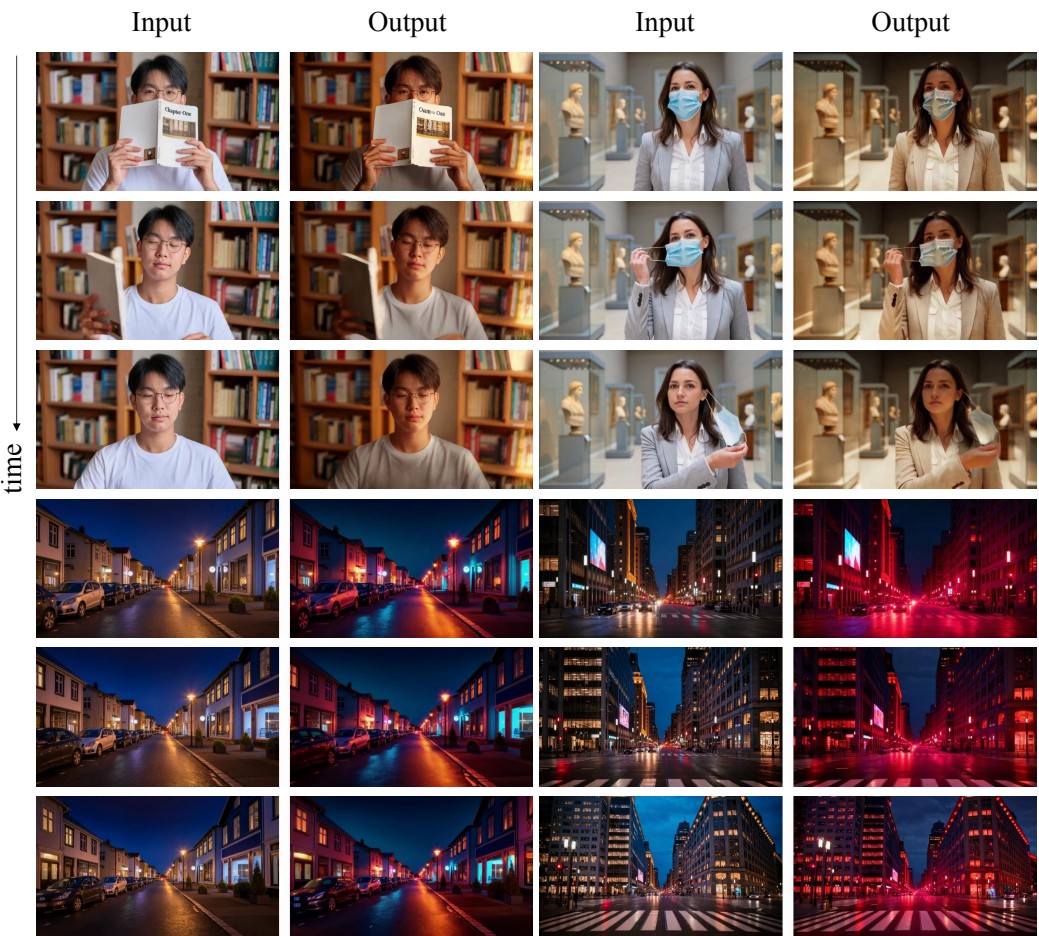

Figure I: Qualitative results showing robustness under occluded reference frames. Light-X maintains coherent illumination propagation even when the reference frame contains partial occlusions.

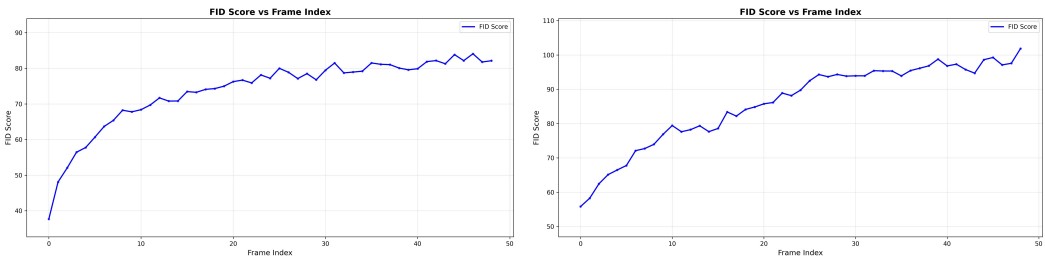

Figure J: FID variation as a function of temporal distance from the reference frame (the first frame). The left part corresponds to the video relighting setting, while the right part corresponds to the joint camera-illumination control setting.

Because synthetic relighting serves only as input rather than ground-truth supervision, its impact is naturally limited. However, the small but consistent gains indicate that generating a larger portion of the training data with Light-X, or exploring iterative self improvement strategies, could still offer potential benefits and represents an interesting direction for future work.

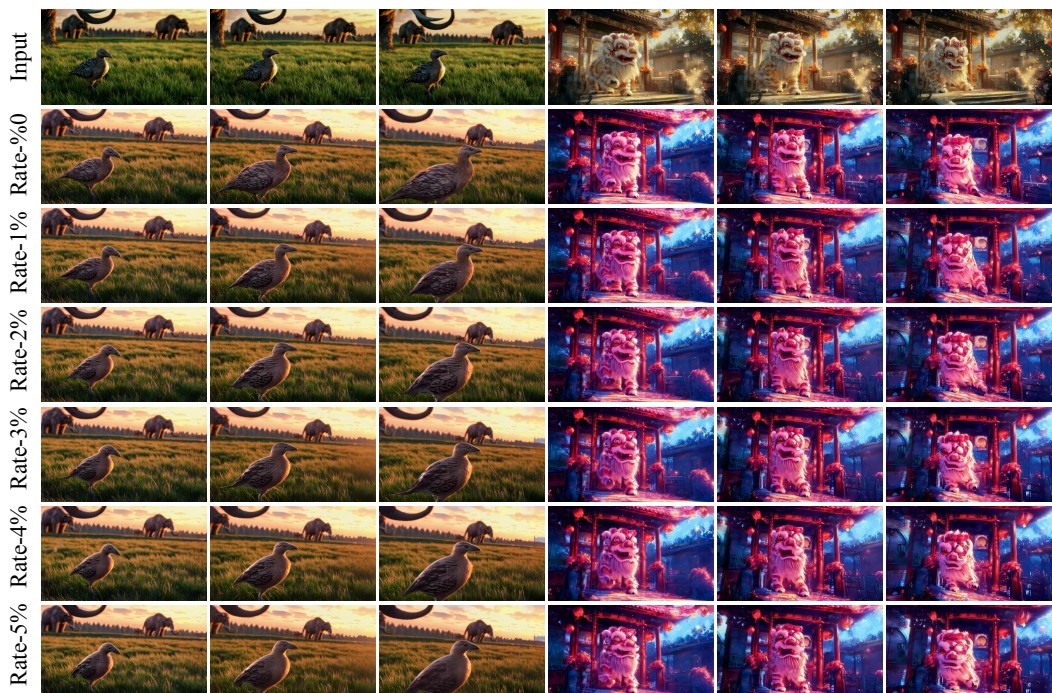

Figure K: Qualitative results under different depth noise levels. Light-X maintains coherent illumination and motion consistency even when depth maps are perturbed with moderate Gaussian noise.

Table H: Performance under increasing depth noise levels.

| Method | FID ↓ | Aesthetic ↑ | Motion Pres. ↓ | CLIP ↑ |
|---|---|---|---|---|
| TC + IC-Light | / | 0.556 | 14.199 | 0.977 |
| TC + LAV | 138.71 | 0.567 | 9.883 | 0.989 |
| LAV + TC | 155.09 | 0.581 | 13.768 | 0.989 |
| TL-Free | 127.77 | 0.585 | 9.533 | 0.990 |
| Ours (rate = 0) | **101.51** | **0.622** | **7.266** | **0.991** |
| Ours (rate = 0.01) | 107.01 | 0.619 | 9.632 | 0.989 |
| Ours (rate = 0.02) | 111.97 | 0.609 | 9.948 | 0.989 |
| Ours (rate = 0.03) | 111.44 | 0.607 | 10.672 | 0.989 |
| Ours (rate = 0.04) | 115.14 | 0.602 | 10.549 | 0.989 |
| Ours (rate = 0.05) | 116.67 | 0.602 | 10.350 | 0.988 |

### D.16 USE OF SYNTHETIC RELIGHTING DATA

We do not adopt synthetic relighting data from graphics engines because such data often fails to capture the complexity and variability of real-world illumination. Our Light-Syn degradation pipeline instead uses in-the-wild videos as ground truth, providing lighting behavior that better matches the model's training requirements. Moreover, IC-Light (Zhang et al., 2025b) already provides strong illumination priors learned from large-scale real and synthetic data, reducing the need for additional synthetic relighting.

To assess whether synthetic data can still bring benefits, we generated 2k synthetic samples with diverse viewpoints and lighting conditions using a DiffusionRenderer-style procedure (Liang et al., 2025). Representative examples are shown in Fig. L. These synthetic samples were mixed with Light-Syn data for training. As reported in Table K, incorporating synthetic relighting consistently degrades performance, suggesting that the domain gap between synthetic and real illumination distributions adversely affects learning. While synthetic relighting offers well-controlled illumination variations, more realistic and diverse synthetic pipelines are needed for it to become truly beneficial.

Table I: Performance under different choices of the relighting reference frame.

| Strategy | FID ↓ | Aesthetic ↑ | Motion Pres. ↓ | CLIP ↑ |
|---|---|---|---|---|
| first | 83.65 | 0.645 | 1.137 | 0.993 |
| mid | 84.85 | 0.634 | 1.249 | 0.993 |
| last | 89.78 | 0.639 | 1.133 | 0.993 |
| random | 85.97 | 0.639 | 1.277 | 0.993 |

Table J: Performance of models fine-tuned with Light-X generated data on the video relighting and joint camera-illumination control tasks.

| Task | Method | FID ↓ | Aesthetic ↑ | Motion Pres. ↓ | CLIP ↑ |
|---|---|---|---|---|---|
| Video Relighting | Ours (original) | 83.65 | 0.645 | 1.137 | 0.993 |
| | Ours (fine-tuned) | 82.00 | 0.643 | 1.134 | 0.993 |
| Joint Cam-Illumination | Ours (original) | 101.06 | 0.623 | 2.007 | 0.989 |
| | Ours (fine-tuned) | 99.35 | 0.622 | 2.171 | 0.989 |

## E   ADDITIONAL ABLATION ANALYSES

Beyond the quantitative ablations presented in Table 6 in the main text and the qualitative comparisons shown in Fig. D, we provide further analysis to better understand the roles of different components, with a focus on the training data composition and the global illumination control module.

### E.1   TRAINING DATA

The three data sources in Light-Syn, including static, dynamic, and AI-generated data, contribute complementary information to the joint camera–illumination control task. Removing static data (a.i) weakens unseen-view synthesis, as static videos provide natural cross-view pairs for stabilizing geometry, as shown in Fig. N. Excluding dynamic data (a.ii) introduces motion artifacts and reduces temporal reliability, as illustrated in Fig. O. Omitting AI-generated data (a.iii) lowers robustness to rare lighting conditions, such as neon or scenes with very bright highlights, where brightness may decay; corresponding qualitative effects are shown in Fig. M. These observations align with the quantitative trends in Table 6 and further demonstrate that the full data mixture helps maintain fidelity, consistency, and stability under diverse lighting.

### E.2   GLOBAL ILLUMINATION CONTROL MODULE

The global illumination control module is crucial for maintaining stable lighting behavior under complex illumination changes. Disabling this module (b.ii) leads to fading or abrupt shifts in brightness, particularly when the scene contains strong directional or spatially varying lighting. With the module enabled, the model is able to preserve coherent global lighting trends, preventing brightness drift and improving temporal consistency. The qualitative results are shown in Fig. P.

Table K: Performance comparison with and without synthetic relighting data on the video relighting and joint camera–illumination control tasks.

| Task | Method | FID ↓ | Aesthetic ↑ | Motion Pres. ↓ | CLIP ↑ |
|---|---|---|---|---|---|
| Video Relighting | Ours (original) | 83.65 | 0.645 | 1.137 | 0.993 |
| | Ours (+ synthetic) | 98.35 | 0.623 | 1.802 | 0.993 |
| Joint Cam-Illumination | Ours (original) | 101.06 | 0.623 | 2.007 | 0.989 |
| | Ours (+ synthetic) | 118.56 | 0.600 | 3.904 | 0.989 |

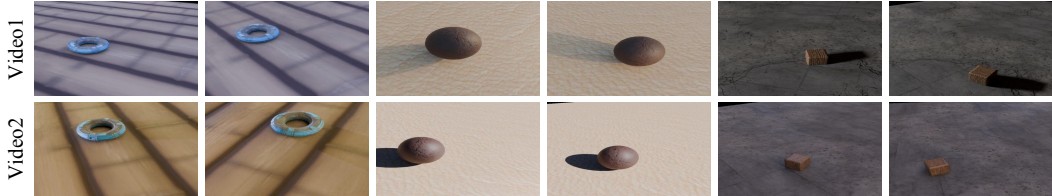

Figure L: Examples of synthetic relighting data generated using graphics engines. These samples exhibit controlled illumination and viewpoint variations.

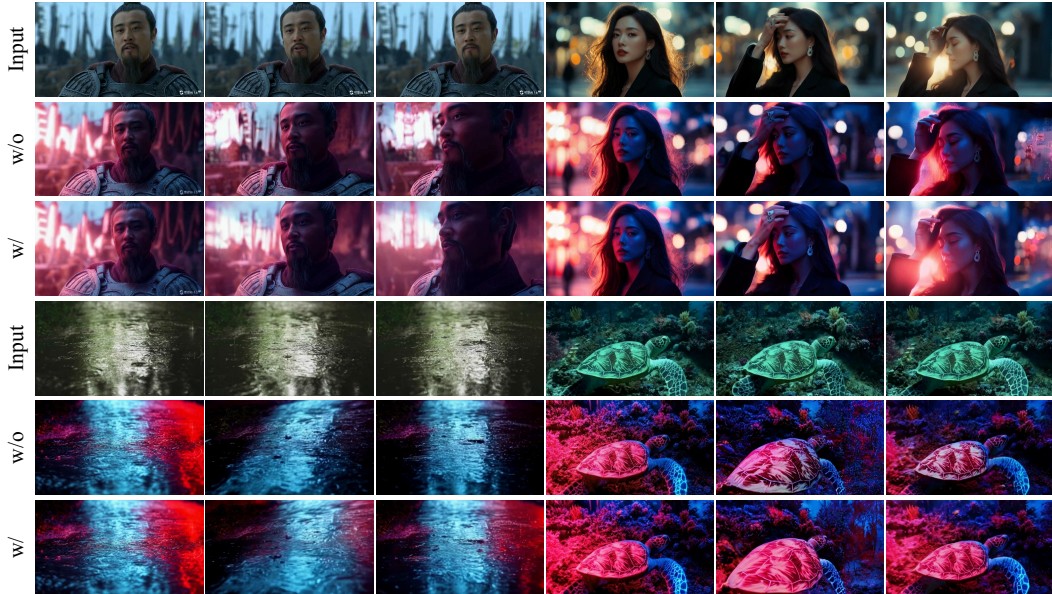

Figure M: Qualitative ablation of the **AI-generated data**.

## F  DISCUSSION

### F.1  LLM USAGE

We acknowledge large language models (LLMs) in the preparation of this manuscript. Specifically, we utilized LLMs for text polishing, grammar correction, and improving the clarity. The core experimental results and scientific contributions remain entirely our own work.

### F.2  ETHICS STATEMENT

Our objective is to advance controllable camera–illumination video generation from monocular videos without introducing additional ethical or safety risks beyond existing generative models. Nevertheless, potential issues such as dataset biases or unintended misuse of generated content cannot be fully excluded. We therefore stress the importance of rigorous data curation, responsible deployment, and transparent reporting to safeguard integrity, fairness, and reproducibility.

### F.3  REPRODUCIBILITY STATEMENT

We emphasize reproducibility by providing detailed descriptions of the proposed framework, including the disentangled design, degradation-based data curation pipeline, and training/evaluation protocols. To support independent validation, we will release the source code, pretrained weights, and curated datasets. This commitment promotes transparency, enables replication, and encourages the research community to extend and improve upon our work in controllable video generation.

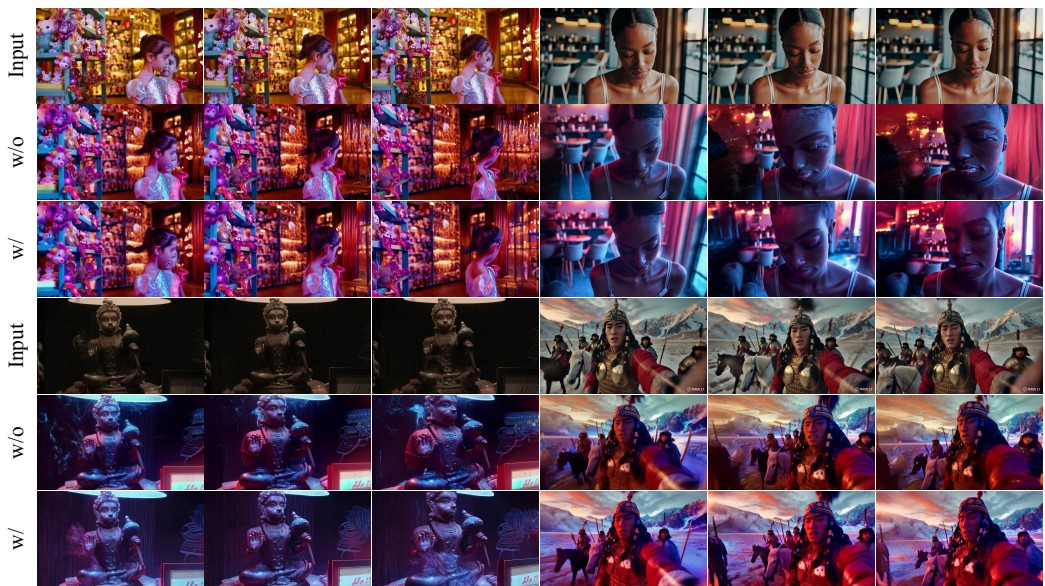

Figure N: Qualitative ablation of the **Static data**.

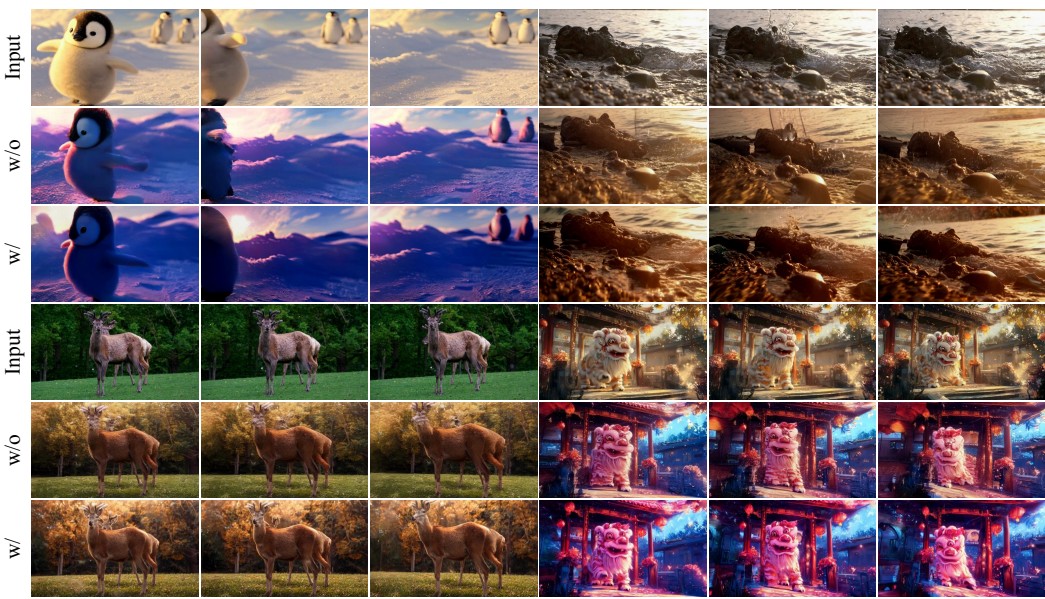

Figure O: Qualitative ablation of the **Dynamic data**.

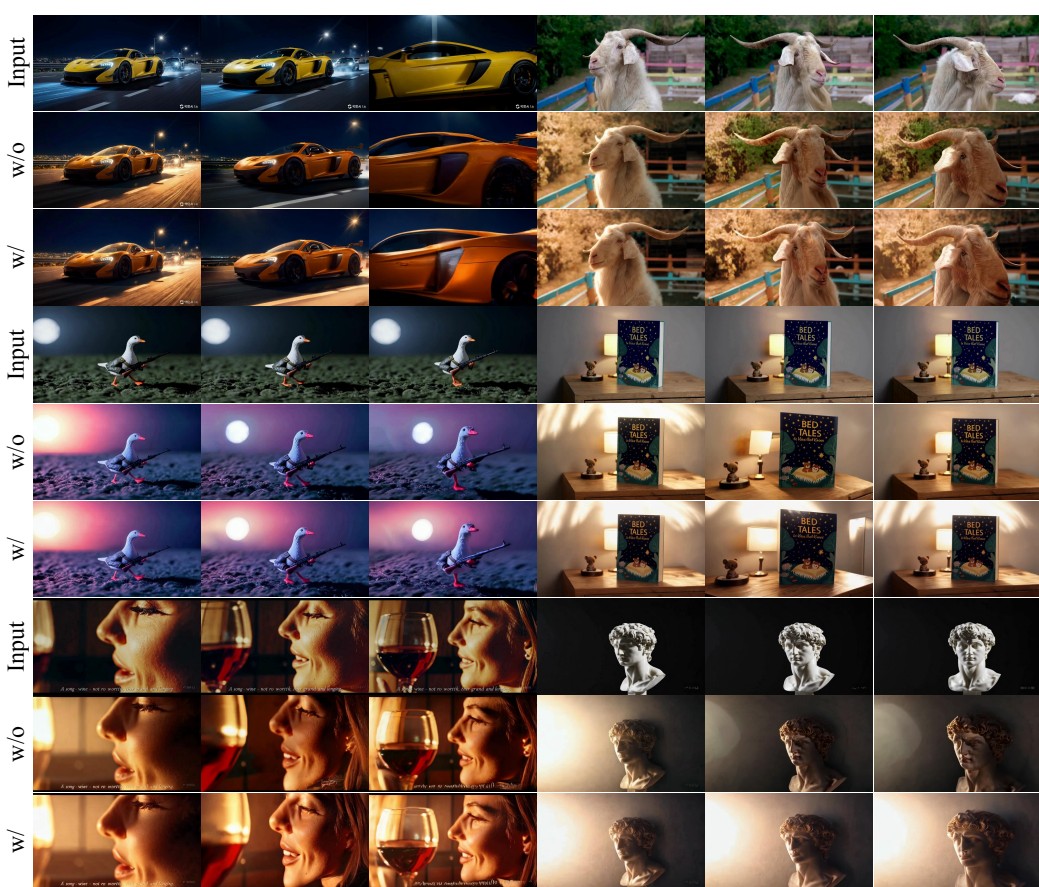

Figure P: Qualitative ablation of the **global illumination control**.

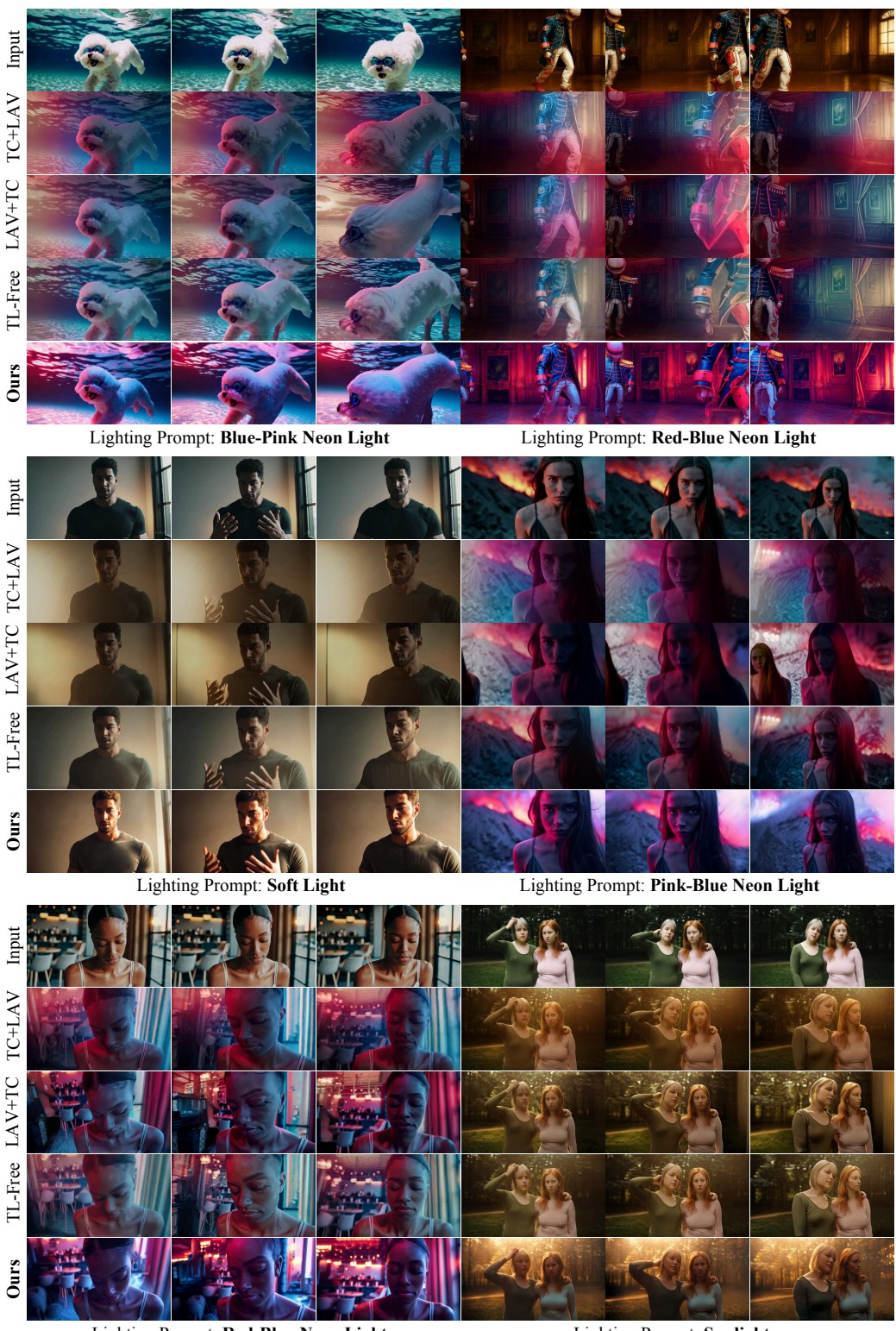

Figure Q: Qualitative comparison of joint camera-illumination control. Our method achieves superior relighting quality, temporal consistency, and realistic novel-view content generation compared to baseline methods.

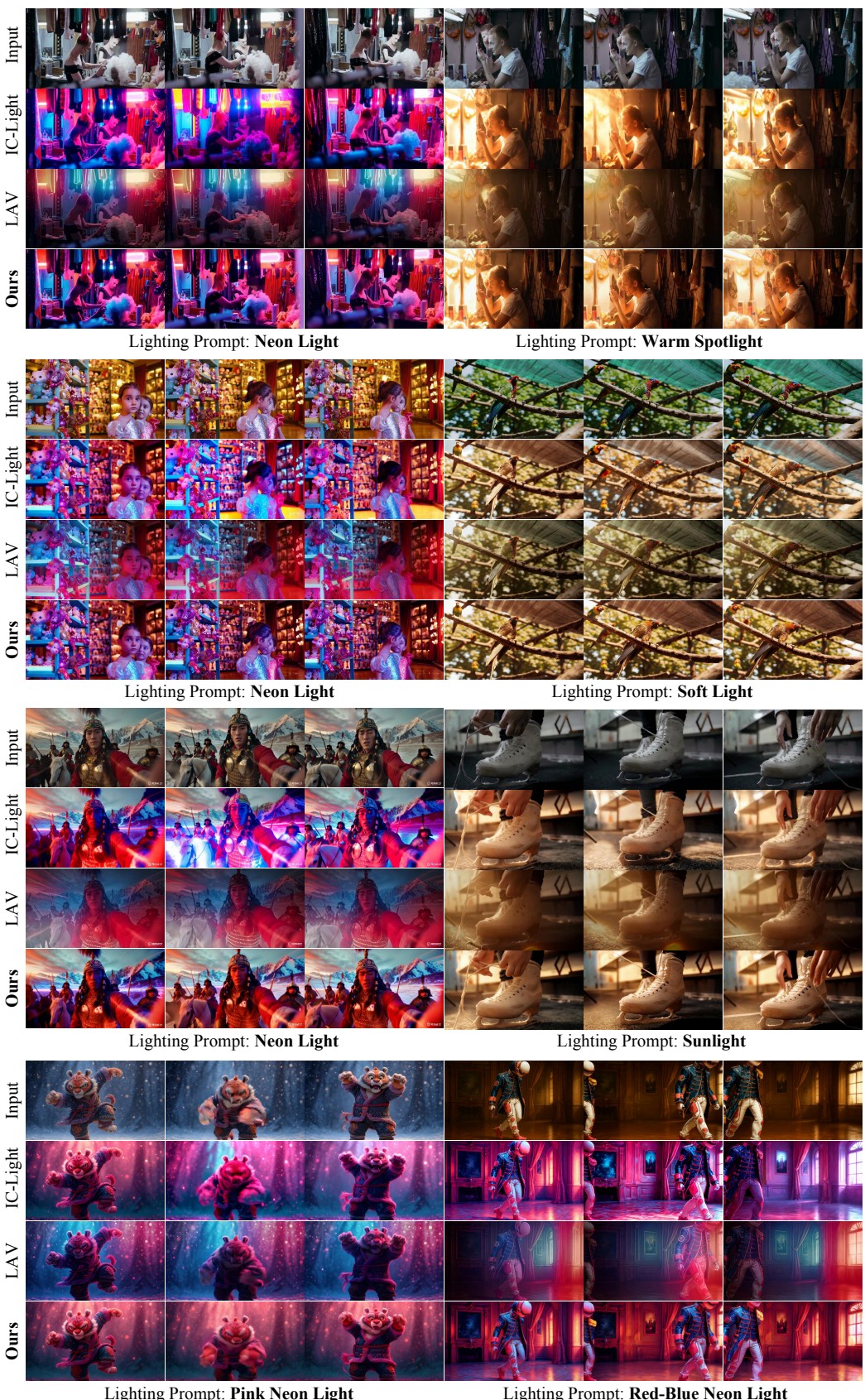

Figure R: Qualitative comparison of text-conditioned video relighting. Our method achieves superior both relighting quality and temporal consistency compared to baseline methods.

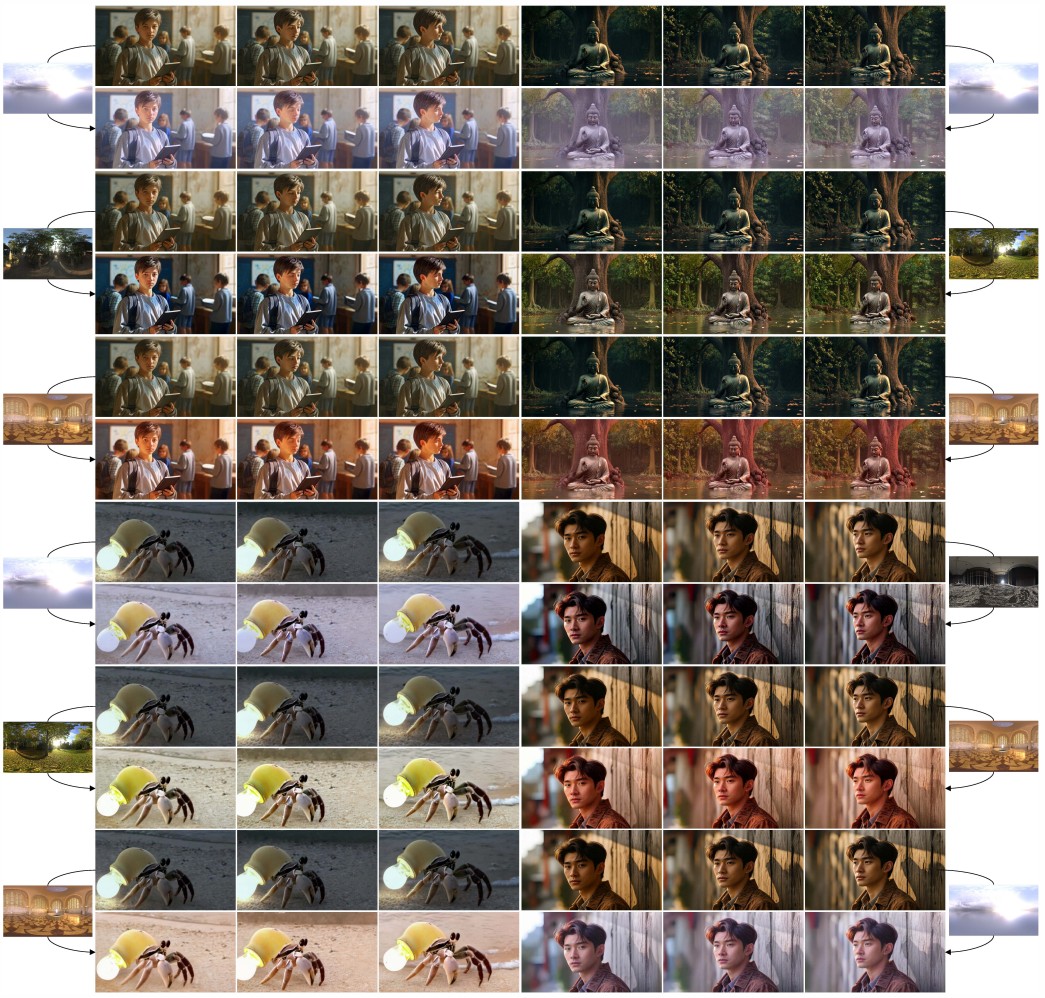

Figure S: Qualitative results of HDR map-conditioned video relighting. Given an input video and an HDR environment map, our model generates a relit video.

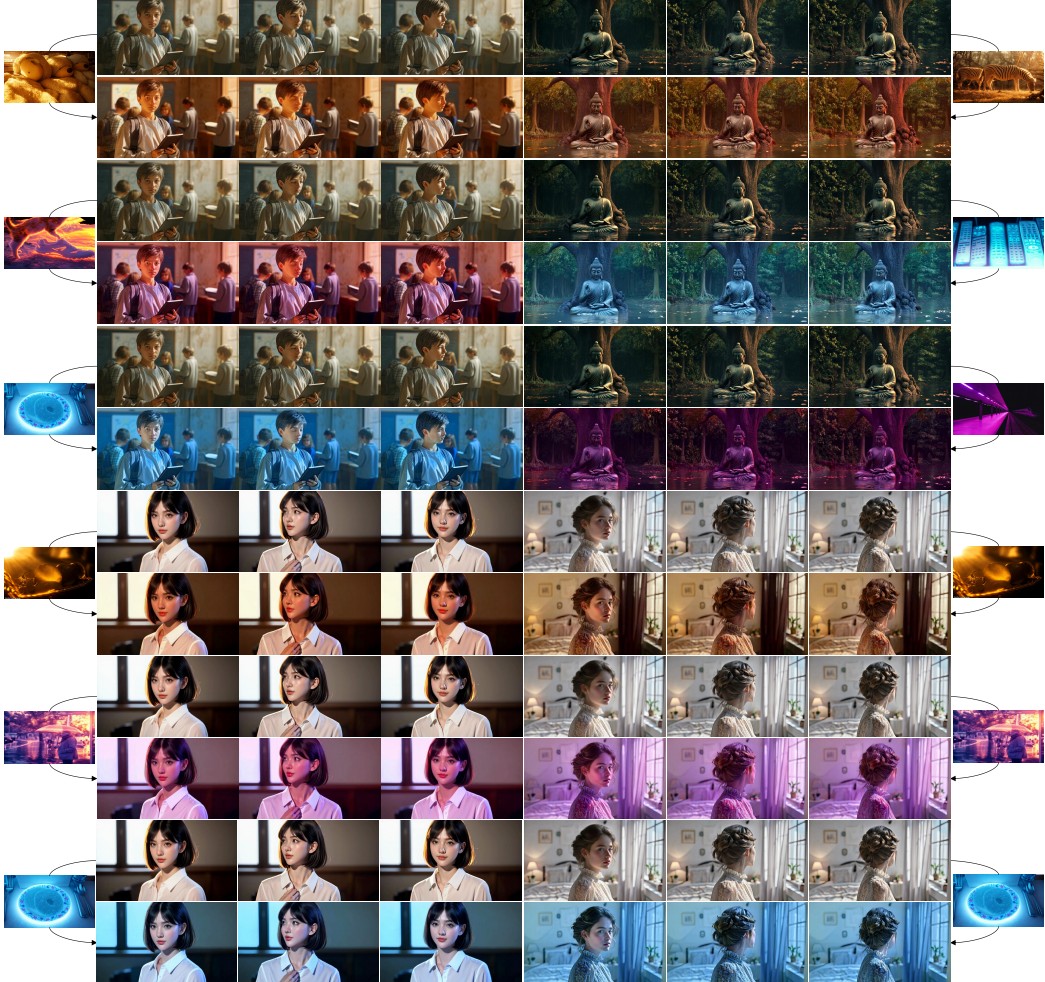

Figure T: Qualitative results of reference image-conditioned video relighting. Here, a reference image provides the target illumination style, which is transferred to the input video while preserving its content and motion.

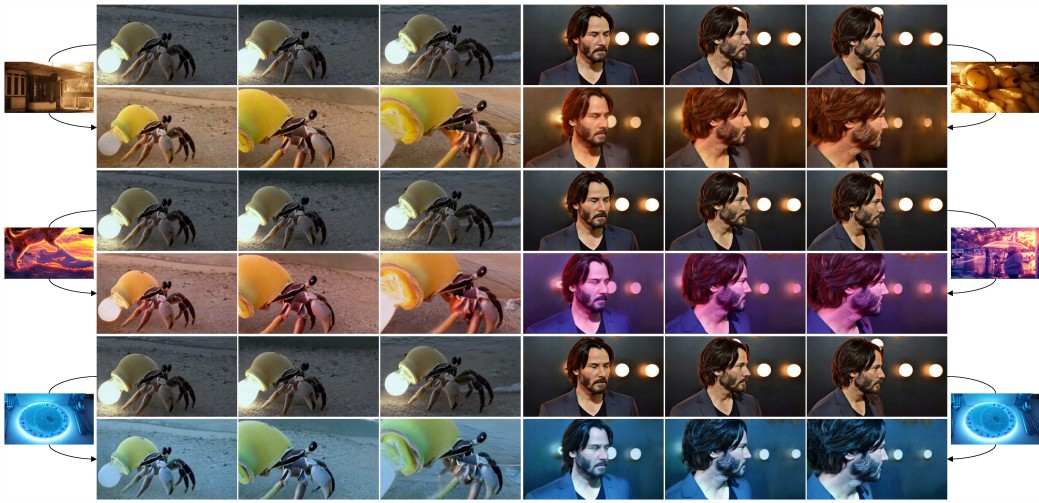

Figure U: Qualitative results of reference image-conditioned joint camera trajectory and illumination control. Here, a reference image provides the target illumination style, which is transferred to the input video while preserving its content and motion.

