# OpenReview forum: "Light-X: Generative 4D Video Rendering with Camera and Illumination Control"
_ICLR.cc/2026/Conference — ICLR 2026 Poster_

### Official Review · Reviewer_BDHW · 2025-10-27

**Soundness:** 3
**Presentation:** 3
**Contribution:** 3
**Rating:** 6
**Confidence:** 3

**Summary:**

This manuscript presents Light-X, a framework achieving end-to-end relighting and redirection given a monocular video. Following TrajectoryCrafter, the proposed architecture retains channel-wise concatenated point-cloud rendering on desired novel camera trajectories and Ref-DiT blocks that incorporates the raw reference video via cross-attention. For illumination control, a frame from the input video is first relighted using IC-Light to obtain a sparse relit video, which is then compressed into a set of illumination tokens via a Q-Former and consumed by a Light-DiT block. The reprojected sparse relit video is also concatenated with the input noise. Additionally, this paper proposes a degradation-based data curation pipeline, Light-Syn, to create training data from in-the-wild videos. Experimental results show that the proposed end-to-end approach outperforms two-stage baselines composed of existing relighting and redirection method on the joint camera–illumination control task, while maintaining flexibility for decoupled control and diverse lighting conditions.

**Strengths:**

1.	The paper is well-structured and easy to follow.
2.	It defines a new task, joint camera-illumination control, which has not been explored by previous works, and designs an end-to-end solution achieving impressive results far superior to naïve combinations of existing methods.
3.	The effort on data curation is commendable.

**Weaknesses:**

1.	Although the method demonstrates clear superiority that direct combination of existing redirection and relighting methods, it is insufficient to convincingly justify the necessity of their end-to-end design. As mentioned in L427, the bottleneck of the TC + LAV baseline lies in the weak relighting capability of the latter. Could it be mitigated by a stronger video-relighting model? Moreover, since the Light-Syn also adopted the TC+LAV approach to produce the source video as shown in Figure C, would it be possible to replace it with the trained Light-X model to generate higher-quality data and then improve the training of Light-X in turn?
2.	The proposed data curation pipeline only relies on the degradation of in-the-wild videos,  potentially introducing a train–inference gap in the distribution of source video and relationship between source and relit videos. For example, the geometry between source and relit videos should be identical in the ideal training data, but TrajectoryCrafter may introduce some deviations. Why not incorporate some accurate synthetic data from graphic engines like IC-Light or RelightVid?

**Questions:**

See weaknesses.

---

> ### Author Response · Authors · 2025-11-24
>
> We sincerely thank Reviewer BDHW for the careful reading of our manuscript and the constructive feedback. We address all concerns below.
>
> **W1.1: Necessity of the End-to-End Design.**
>
> Our end-to-end design is necessary for three reasons.
> 1) Stronger relighting does not solve the two-stage limitation. Even when replacing LAV with TC-Light (NeurIPS 2025), the most recent illumination control method to our knowledge, this sequential pipeline still lags behind Light-X, as shown in the table below.
> 2) Efficiency. Two-stage pipelines run relighting and redirection sequentially, whereas Light-X performs both in a single pass.
> 3) Geometry–illumination interaction.
> Illumination depends on camera motion (e.g., shadows, specularities, occlusions), but two-step pipelines cannot model this dependency because relighting and redirection are performed separately. Light-X learns geometry and lighting jointly, leading to more coherent results.
>
> | Method        | FID ↓    | Aesthetic ↑ | Motion Pres. ↓ | CLIP ↑ |
> |---------------|----------|-------------|----------------|--------|
> | TC + IC-Light | /        | 0.573       | 6.558          | 0.976  |
> | TC + LAV      | 138.89   | 0.574       | 4.327          | 0.986  |
> | LAV + TC      | 144.61   | 0.596       | 5.027          | 0.987  |
> | TL-Free       | 122.73   | 0.595       | 3.356          | 0.987  |
> | TC + TC-Light | 154.99   | 0.534       | 4.276          | 0.986  |
> | TC-Light + TC | 161.76   | 0.555       | 5.563          | 0.988  |
> | Ours          | **101.06** | **0.623**   | **2.007**      | **0.989** |
>
>
> **W1.2: Using Light-X Itself for Data Generation.**
>
> We thank the reviewer for this interesting and valuable idea.
> Indeed, using Light-X to generate higher-quality relit videos is appealing, since Light-X produces more consistent illumination and more realistic propagation than existing relighting models.
> To validate this idea, we conducted a preliminary experiment:
> we randomly selected 2k samples from our training set, relit them using Light-X, and then fine-tuned the original model using these Light-X-generated data. The comparison between the original and the fine-tuned model is shown below:
>
> The results for video relighting are shown in the table below:
>
> | Method           | FID ↓ | Aesthetic ↑ | Motion Pres. ↓ | CLIP ↑ |
> |------------------|-------|-------------|----------------|--------|
> | Ours (original)  | 83.65 | 0.645       | 1.137          | 0.993  |
> | Ours (fine-tuned)| 82.00 | 0.643       | 1.134          | 0.993  |
>
>
> The results for joint camera–illumination control are shown in the table below:
>
> | Method           | FID ↓  | Aesthetic ↑ | Motion Pres. ↓ | CLIP ↑ |
> |------------------|--------|-------------|----------------|--------|
> | Ours (original)  | 101.06 | 0.623       | 2.007          | 0.989  |
> | Ours (fine-tuned)| 99.35  | 0.622       | 2.171          | 0.989  |
>
>
> The fine-tuned model using a subset of the data shows a slight improvement in some metrics, suggesting that higher-quality data can indeed provide certain benefits.
> However, the gain is modest, suggesting that the original data is already sufficiently effective.
> This is consistent with the training data requirements described in our Detailed Data Curation section. Although prior methods may produce imperfect relighting (e.g., temporal inconsistency), these relit frames serve only as inputs rather than supervision. The model is still trained against real in-the-wild videos, which mitigates the impact of imperfections in the synthetic relighting.
>
> Overall, we find the reviewer’s suggestion highly insightful. We plan to explore a full Light X based data regeneration pipeline, relighting the entire training set and retraining the model, to more thoroughly validate its potential in future work.

---

> > ### Author Response · Authors · 2025-11-24
> >
> > **W2: Use of Synthetic Relighting Data.**
> >
> > We thank the reviewer for this thoughtful and insightful comment. We chose not to rely on synthetic relighting data from graphics engines for the following reasons:
> > 1) IC-Light already provides strong relighting priors. IC-Light is trained on large-scale synthetic and real data and already provides reliable illumination cues, which Light-X uses as lighting priors to propagate across frames and viewpoints.
> > 2) Our Light-Syn pipeline matches the training requirements.
> > As detailed in the Data Curation section, our degradation-based pipeline is designed to meet the training data requirements. Using in-the-wild videos as ground truth enables the model to learn real-world illumination behavior, which synthetic data cannot fully capture.
> > 3) Synthetic data may introduce a domain gap.
> > Following the reviewer’s suggestion, we generated 2k synthetic samples with varying camera viewpoints and lighting conditions, using a DiffusionRenderer-style data generation procedure (see Fig.L for sample visualizations).
> > When these synthetic samples were mixed with Light-Syn data for training, we observed a performance drop, likely caused by the domain gap between synthetic and real-world illumination distributions.
> >
> >
> > The results for video relighting are shown in the table below:
> >
> > | Method            | FID ↓ | Aesthetic ↑ | Motion Pres. ↓ | CLIP ↑ |
> > |-------------------|-------|-------------|----------------|--------|
> > | Ours (original)   | 83.65 | 0.645       | 1.137          | 0.993  |
> > | Ours (+synthetic) | 98.35 | 0.623       | 1.802          | 0.993  |
> >
> >
> > The results for joint camera–illumination control are shown in the table below:
> >
> > | Method             | FID ↓  | Aesthetic ↑ | Motion Pres. ↓ | CLIP ↑ |
> > |--------------------|--------|-------------|----------------|--------|
> > | Ours (original)    | 101.06 | 0.623       | 2.007          | 0.989  |
> > | Ours (+synthetic)  | 118.56 | 0.600       | 3.904          | 0.989  |
> >
> >
> > While synthetic relighting offers a smaller training-inference gap, it also introduces a noticeable domain gap. Our results show that this domain mismatch currently outweighs its potential benefits. Exploring more advanced synthetic pipelines capable of producing more realistic and complex paired relighting data remains a promising direction for future work.

---

> > > ### Comment · Reviewer_BDHW · 2025-11-27
> > >
> > > Thanks for the author’s efforts in addressing my concern and question. I am satisfied with the additional results which provide a convincing justification for their end-to-end design and degradation-based pipeline. Concretely, the quantitative results in the first table suggest that the performance achieved by the proposed single-pass pipeline cannot be simply obtained by employing a stronger relighting method in the two-stage solution. I also agree with the author’s arguments about its efficiency advantage. Furthermore, I appreciate their preliminary exploration of bootstrap training, and acknowledge that both its result and the last two tables indicate that the degradation-based data curation pipeline is good enough. Therefore, I am willing to raise my rating to 8.

---

> > > > ### Author Response · Authors · 2025-11-27
> > > >
> > > > We sincerely thank Reviewer BDHW for the encouraging follow-up comments and for the willingness to raise the rating to 8. We truly appreciate the reviewer’s positive recognition of the end-to-end design, efficiency advantages, and the effectiveness of the degradation-based data curation pipeline. We also thank the reviewer for acknowledging our preliminary bootstrap-training exploration. The reviewer’s constructive suggestions and thoughtful engagement have greatly helped improve the clarity and completeness of the paper.

---

### Official Review · Reviewer_LGn4 · 2025-10-29

**Soundness:** 2
**Presentation:** 3
**Contribution:** 3
**Rating:** 6
**Confidence:** 3

**Summary:**

The paper proposes Light-X, a video diffusion framework that enables joint control of camera trajectory and illumination from a monocular input video. The method explicitly decouples geometry/motion from lighting: dynamic point clouds projected along user-specified camera paths provide geometric cues, while a single relit frame is re-projected through the same geometry to supply illumination cues. To obtain training pairs without multi-view or multi-illumination captures, the authors introduce Light-Syn, a degradation pipeline that turns in-the-wild videos into paired inputs/targets and aligned conditioning renders/masks. Experiments show improvements over composed baselines for the new joint camera+illumination task and competitive results for video relighting under text and background conditions, with extensive ablations.

**Strengths:**

- The proposed factorization method supplies the model with fine-grained, geometry-aligned cues (projected source views/masks and projected relit views/masks) and complementary global illumination tokens (Q-Former), which are technically sound and easy to reason about.
- For the new joint control task (no direct prior), the paper composes reasonable baselines from camera-control and relighting methods, and also introduces a tailored training-free baseline with documented adaptations.
- The evaluation method and metrics seem solid. The authors fix random seeds over prompts, light directions, and trajectories across methods; metrics cover fidelity (FID and Aesthetic Preference) and temporal aspects (CLIP similarity across frames, Motion Preservation via RAFT). A 57-person user study evaluates Relighting Quality, Video Smoothness, ID Preservation, and 4D consistency.
- Ablations show the benefits of different design choices proposed in the paper.

**Weaknesses:**

- For joint control, FID is computed against IC-Light-relit outputs on a TrajectoryCrafter sequence. That anchors “ground truth look” to IC-Light’s aesthetics and may bias the metric toward that method’s style.
- The related work mentions several recent camera/lighting control or camera-controlled generators (e.g., VidCraft3, ReCamMaster, CAMI2V, Free4D/VD3D) that appear not to be included in quantitative comparisons. A brief justification or attempts to compare to the most recent DiT-based camera-control systems (or to lighting-control contemporaries) would strengthen claims of SOTA.
- Light-Syn mixes static, dynamic, and AI-generated sources. While this increases diversity, it risks domain confounds (e.g., the model learning priors from GenAI models). Cross-domain breakdowns (real-only vs AI-only) would clarify generalization.

**Questions:**

In addition to the points listed in the weakness section, I have some additional questions:

- The method relights only one frame to build lighting cues, then relies on Light-DiT for global consistency. It would help to quantify how performance drops as temporal distance grows, or when the relit frame has occlusions/poor depth.
- How robust is Light-X to camera intrinsics errors and depth noise? A small controlled study would strengthen the geometric claims.
- What motivates using the first frame as the relighting frame? Is there a better way of choosing a reference frame from a video, as sometimes the first frame might not be the most informative?

---

> ### Author Response · Authors · 2025-11-24
>
> We sincerely thank Reviewer LGn4 for the detailed and constructive review, as well as the insightful questions and suggestions. We address all concerns below.
>
> **W1: Potential bias of using IC-Light as the FID reference.**
>
> The reviewer raises a valuable point regarding the potential bias introduced by using IC-Light as the FID reference.
>
> 1. **Bias acknowledged.**
>    We agree that IC-Light-based FID has inherent bias. We follow LAV’s protocol due to the lack of real paired relighting data.
>
> 2. **Why the comparison remains fair.**
>    All baselines inherit IC-Light behavior (LAV directly combines IC-Light while RelightVid fine-tunes it), so this bias affects all methods and does not favor ours.
>
> 3. **User study** also confirms better lighting quality.
>
> 4. **Additional unbiased evaluation.**
> We conduct a real-video degradation experiment to obtain an unbiased, IC-Light-free evaluation.
> Specifically, we use in-the-wild real videos as ground truth, estimate their lighting prompts using LLaVA, and generate a *degraded version* using LAV under a neutral lighting prompt.
> During testing, the degraded video serves as the input, and the LLaVA-derived lighting prompt is used as the illumination condition.
> We then evaluate the relit outputs directly against the real videos using standard metrics (PSNR/SSIM/LPIPS/FVD) on 200 videos.
> Our method achieves the best scores across all metrics, further demonstrating its superior relighting quality and temporal consistency.
>
>
> The results for video relighting are shown in the table below:
>
> | Method   | PSNR ↑ | SSIM ↑ | LPIPS ↓ | FVD ↓ |
> |----------|--------|--------|---------|-------|
> | IC-Light | 11.75  | 0.517  | 0.422   | 67.50 |
> | LAV      | 12.66  | 0.530  | 0.429   | 74.64 |
> | Ours     | **13.84** | **0.581** | **0.369** | **56.60** |
>
>
> The results for joint camera–illumination control are shown in the table below:
>
> | Method      | PSNR ↑  | SSIM ↑  | LPIPS ↓ | FVD ↓   |
> |-------------|---------|---------|---------|---------|
> | TC + IC-Light | 10.963  | 0.4557  | 0.4744  | 58.8538 |
> | TC + LAV    | 12.178  | 0.4702  | 0.5082  | 73.7790 |
> | LAV + TC    | 12.476  | 0.4626  | 0.4793  | 60.9497 |
> | TL-Free     | 13.486  | 0.5466  | 0.4180  | 54.4410 |
> | Ours        | **13.955** | **0.5819** | **0.3777** | **45.9116** |
>
>
>
> **W2: Comparison with more recent camera- or lighting-controlled methods.**
>
> We appreciate the reviewer’s suggestion. In the revision, we additionally compare Light-X against recent representative systems from both categories: **i) Camera control:** **ReCamMaster** (ICCV 2025), which does not rely on explicit 3D representations, and **Free4D** (ICCV 2025), which incorporates an explicit 3D representation. **ii) Lighting control:** **TC-Light** (NeurIPS 2025), the most recently published illumination-control method to our knowledge. Light-X maintains superior image fidelity, aesthetic quality, and motion consistency even against these recent systems, further supporting its SOTA performance.
>
> The results for joint camera–illumination control are shown in the table below:
>
> | Method           | FID ↓    | Aesthetic ↑ | Motion Pres. ↓ | CLIP ↑ |
> |------------------|----------|-------------|----------------|--------|
> | TC + IC-Light    |    /     | 0.573       | 6.558          | 0.976  |
> | TC + LAV         | 138.89   | 0.574       | 4.327          | 0.986  |
> | LAV + TC         | 144.61   | 0.596       | 5.027          | 0.987  |
> | TL-Free          | 122.73   | 0.595       | 3.356          | 0.987  |
> | ReCam + IC-Light |   /      | 0.513       | 6.511          | 0.973  |
> | LAV + ReCam      | 163.56   | 0.514       | 7.259          | **0.989** |
> | ReCam + LAV      | 152.03   | 0.501       | 3.157          | 0.987  |
> | TC + TC-Light    | 154.99   | 0.534       | 4.276          | 0.986  |
> | TC-Light + TC    | 161.76   | 0.555       | 5.563          | 0.988  |
> | Ours             | **101.06** | **0.623**   | **2.007**      | **0.989** |
>
>
>
> The results for video relighting are shown in the table below:
>
> | Method    | FID ↓  | Aesthetic ↑ | Motion Pres. ↓ | CLIP ↑ |
> |-----------|--------|-------------|----------------|--------|
> | IC-Light  |   /    | 0.632       | 3.293          | 0.983  |
> | LAV       | 112.45 | 0.614       | 2.115          | 0.991  |
> | TC-Light  | 144.32 | 0.546       | 1.657          | 0.991  |
> | Ours      | **83.65** | **0.645**   | **1.137**      | **0.993** |
>
>
> For Free4D, since it requires per-scene optimization (typically taking over an hour per scene), we evaluate it on the 10 scenes provided on the project page for convenience. The results on these scenes are reported below.
>
> | Method           | FID ↓  | Aesthetic ↑ | Motion Pres. ↓ | CLIP ↑ |
> |------------------|--------|-------------|----------------|--------|
> | Free4D + IC-Light |   /    | 0.576       | 0.823          | 0.990  |
> | Free4D + LAV    | 98.85  | 0.574       | 0.549          | 0.996  |
> | Ours            | **73.98** | **0.583**   | **0.349**      | **0.997** |

---

> > ### Author Response · Authors · 2025-11-24
> >
> > **W3: Data Source Ablation.**
> >
> > We appreciate the reviewer’s insightful comment on potential domain bias.
> > Table 4 provides a cross-domain breakdown by removing each data source: static (a.i), dynamic (a.ii), and AI-generated data (a.iii, i.e., real-only). Performance drops in all three settings, showing that static data provides cross-view cues, dynamic data provides motion cues, and AI-generated data improves robustness to rare lighting. This indicates that Light-Syn benefits from complementary domains rather than overfitting to any single one.
> > The corresponding visual comparisons are shown in **Fig.M**, **Fig.N**, and **Fig.O**, and additional ablation videos are included in the supplementary material for more intuitive comparison (**LGn4-W3-Data-Ablation.mp4**).
> >
> > **Q1.1: FID Degradation with Temporal Distance.**
> >
> > Thanks for this insightful question. We quantify how performance changes with temporal distance from the relit frame, with the corresponding curves shown in **Fig.J** of the revised manuscript. As expected, FID increases gradually because it is computed against the IC-Light–relit reference: the first frame is directly relit by IC-Light, whereas later frames rely on Light-X to propagate illumination cues. For video relighting only, FID increases smoothly from 38 to 82 over 49 frames.
> > For camera and relighting joint control, FID rises from 56 to 100. Importantly, even the final frame still surpasses baseline methods such as LAV (FID: 112.45) and TL-Free (FID: 122.73). These results demonstrate that Light-X maintains competitive quality even at large temporal distances from the reference frame.
> >
> > **Q1.2: Performance with Occluded Relit Frames.**
> >
> > Thanks for the reviewer’s constructive question. Light-X remains robust even when the relit reference frame (using the first frame as an example) is partially occluded or contains incomplete scene information. We provide the corresponding visual results in **Fig.I** of the revised manuscript and in the supplementary video (**LGn4-Q1.2-Reference-Occlusion.mp4**). Illumination cues are still propagated coherently even when the reference frame includes occlusions (e.g., a book or mask covering part of the face). Similarly, in zoom-out scenarios where later frames reveal new regions, Light-X still produces reasonable relighting for newly visible areas.
> >
> >
> > **Q1.3\&Q2: Robustness to camera intrinsics errors and depth noise.**
> >
> > Thanks for the reviewer’s valuable suggestion.
> > 1) Camera intrinsics. In practice, obtaining accurate intrinsics requires a separate calibration procedure. For simplicity and efficiency, we therefore use a fixed intrinsics setting (focal length = 500) for all scenes during inference. Of course, using the true intrinsics would be ideal.
> >
> > 2) Depth noise. Light-X relies on projected point-cloud views as soft geometric cues, so inaccurate depth naturally introduces biased geometry and may affect performance. However, the method does not require highly accurate depth and is reasonably robust to moderate estimation errors. Following the reviewer’s suggestion, we conducted a small controlled study on 12 randomly selected scenes. Using DepthCrafter as the default depth estimator, we injected Gaussian noise into the depth maps:
> > $\tilde{D} = D + \epsilon \cdot D,\quad \epsilon \sim \mathcal{N}(0,\ \text{rate}),$
> > where rate controls the perturbation level. The performance of Light-X under different noise levels and baseline methods is reported in the table below, showing that Light-X degrades gracefully and consistently outperforms all baselines.We additionally include the corresponding visual results in **Fig.K** and the supplementary video (**LGn4-Q1.3\&Q2-Depth-Noise.mp4**), offering a more intuitive comparison.
> >
> >
> > | Method               | FID (↓) | Aesthetic (↑) | Motion Pres. (↓) | CLIP (↑) |
> > |----------------------|---------|---------------|------------------|----------|
> > | TC + IC-Light        | /       | 0.556         | 14.199           | 0.977    |
> > | TC + LAV             | 138.71  | 0.567         | 9.883            | 0.989    |
> > | LAV + TC             | 155.09  | 0.581         | 13.768           | 0.989    |
> > | TL-Free              | 127.77  | 0.585         | 9.533            | 0.990    |
> > | Ours (Noise = 0)     | **101.51** | **0.622**    | **7.266**        | **0.991** |
> > | Ours (Noise = 0.01)  | 107.01  | 0.619         | 9.632            | 0.989    |
> > | Ours (Noise = 0.02)  | 111.97  | 0.609         | 9.948            | 0.989    |
> > | Ours (Noise = 0.03)  | 111.44  | 0.607         | 10.672           | 0.989    |
> > | Ours (Noise = 0.04)  | 115.14  | 0.602         | 10.549           | 0.989    |
> > | Ours (Noise = 0.05)  | 116.67  | 0.602         | 10.350           | 0.988    |

---

> > > ### Author Response · Authors · 2025-11-24
> > >
> > > **Q3: Choice of the Relighting Reference Frame.**
> > >
> > > We thank the reviewer for the thoughtful question.
> > > Light-X does not rely on using the first frame as the relighting reference.
> > > During training, the relit frame is randomly sampled, and during inference any frame can be used.
> > > In the main paper, we adopt the first frame purely for implementation convenience and reproducibility.
> > > To assess robustness, we evaluated four reference-frame strategies (“first”, “middle”, “last”, “random”) on 200 videos. Light-X achieves similar performance across all choices, demonstrating that Light-X is robust to the choice of reference frame,
> > >
> > >
> > > | Strategy | FID (↓) | Aesthetic (↑) | Motion Pres. (↓) | CLIP (↑) |
> > > |----------|---------|---------------|------------------|----------|
> > > | first    | 83.65   | 0.645         | 1.137            | 0.993    |
> > > | mid      | 84.85   | 0.634         | 1.249            | 0.993    |
> > > | last     | 89.78   | 0.639         | 1.133            | 0.993    |
> > > | random   | 85.97   | 0.639         | 1.277            | 0.993    |
> > >
> > >
> > > Regarding the reviewer’s point about whether there may be a better way to choose the reference frame, we agree that this is an interesting direction. In general, a reference frame that contains more complete scene information could make illumination propagation easier and more reliable. Developing an automatic strategy to select such an informative frame is a promising extension, which we leave for future work.

---

### Official Review · Reviewer_JNPV · 2025-11-01

**Soundness:** 3
**Presentation:** 4
**Contribution:** 4
**Rating:** 8
**Confidence:** 3

**Summary:**

This paper adds joint control of camera trajectory and illumination to render from monocular videos. They use a disentangled design to decouple geometry and lighting. They also use a degradation-based pipeline with inverse-mapping to synthesizes training pairs.
a clear understanding of the limitations of prior approaches, a well-defined methodology with ablations for every aspect, and valid baselines for evaluation. Starting with an input video, two sets of inputs for diffusion are generated: Camera control and Illumination control. These priors are used as conditions for video diffusion, passed through DiT blocks for denoising. They also introduce a Light-DiT layer for global illumination control as they observe that illumination strength diminishes as frames move further away from the relit frame. Experiments are sound and the results are very good.

**Strengths:**

- First paper to tackle the novel problem of video generation with joint camera and illumination control for monocular videos by providing conditioning to a diffusion transformer.
- The Light-Syn pipeline uses an effective degradation idea for training data creation. The data sources comprise static scenes, dynamic scenes, and AI-generated videos with ablations justifying the significance of each source.
- Light-DiT layer allows global illumination control by using a Q-Former to prevent diminishing illumination strength for frames that are further away from the relit frame.
- Paper is well written and easy to follow.

**Weaknesses:**

No major weakness.

Minor Weakness: The evaluation metric using FID between the output image and IC-Light would have a potential evaluation bias, as the model is judged on its ability to mimic the behavior of a component (IC-Light) used in its own conditioning scheme.

**Questions:**

How would the model perform for camera trajectories significantly different from the source? The DiT blocks would need to in-paint a large area with little conditioning, and would errors in this propagate to the next frames?

---

> ### Author Response · Authors · 2025-11-24
>
> We sincerely thank Reviewer JNPV for the positive and encouraging review, as well as the insightful comments. We address all concerns below.
>
> **W1: Potential bias of using IC-Light as the FID reference.**
>
> The reviewer raises a valuable point regarding the potential bias introduced by using IC-Light as the FID reference.
>
> 1. **Bias acknowledged.**
>    We agree that IC-Light-based FID has inherent bias. We follow LAV’s protocol due to the lack of real paired relighting data.
>
> 2. **Why the comparison remains fair.**
>    All baselines inherit IC-Light behavior (LAV directly combines IC-Light while RelightVid fine-tunes it), so this bias affects all methods and does not favor ours.
>
> 3. **User study** also confirms better lighting quality.
>
> 4. **Additional unbiased evaluation.**
> We conduct a real-video degradation experiment to obtain an unbiased, IC-Light-free evaluation.
> Specifically, we use in-the-wild real videos as ground truth, estimate their lighting prompts using LLaVA, and generate a *degraded version* using LAV under a neutral lighting prompt.
> During testing, the degraded video serves as the input, and the LLaVA-derived lighting prompt is used as the illumination condition.
> We then evaluate the relit outputs directly against the real videos using standard metrics (PSNR/SSIM/LPIPS/FVD) on 200 videos.
> Our method achieves the best scores across all metrics, further demonstrating its superior relighting quality and temporal consistency.
>
> The results for video relighting are shown in the table below:
>
> | Method   | PSNR ↑ | SSIM ↑ | LPIPS ↓ | FVD ↓ |
> |----------|--------|--------|---------|-------|
> | IC-Light | 11.75  | 0.517  | 0.422   | 67.50 |
> | LAV      | 12.66  | 0.530  | 0.429   | 74.64 |
> | Ours     | **13.84** | **0.581** | **0.369** | **56.60** |
>
>
> The results for joint camera–illumination control are shown in the table below:
>
> | Method      | PSNR ↑  | SSIM ↑  | LPIPS ↓ | FVD ↓   |
> |-------------|---------|---------|---------|---------|
> | TC + IC-Light | 10.963  | 0.4557  | 0.4744  | 58.8538 |
> | TC + LAV    | 12.178  | 0.4702  | 0.5082  | 73.7790 |
> | LAV + TC    | 12.476  | 0.4626  | 0.4793  | 60.9497 |
> | TL-Free     | 13.486  | 0.5466  | 0.4180  | 54.4410 |
> | Ours        | **13.955** | **0.5819** | **0.3777** | **45.9116** |
>
>
> **Q1: Large Camera Trajectories.**
>
> The reviewer raises an insightful concern. As also discussed in our Limitations and Future Work section, and similar to other methods that condition on point clouds, Light-X relies on point-cloud priors. Under extremely wide camera motions, the available point-cloud cues become very limited or even absent, which can adversely affect the generation quality. Nonetheless, the model demonstrates robustness to substantial viewpoint deviations (approximately up to 60°), as demonstrated in **Fig.H** of the revised manuscript and in the supplementary video (**JNPV-Q1-Large-Camera.mp4**).

---

### Official Review · Reviewer_6GDF · 2025-11-01

**Soundness:** 3
**Presentation:** 3
**Contribution:** 2
**Rating:** 6
**Confidence:** 4

**Summary:**

The work presents a method, Light-X, for generative 4D video rendering, enabling the disentanglement of scene geometry and illumination in dynamic scenes. Light-X allows for controllable video relighting and camera trajectory redirection from monocular video inputs. Experiments show that the introduced model successfully achieves visually impressive relighting and redirection results across various tests.

**Strengths:**

* Light-X aims to achieve the very challenging task of disentangling geometry and illumination for dynamic scenes, yet it manages to achieve quite impressive visual results, as shown in the provided video.

* This work also introduces an easy-to-setup data curation pipeline for creating paired original and relighted videos of geometrically coherent dynamic scenes.

* The manuscript is well-structured and easy to follow.

**Weaknesses:**

* **Limited technical contribution within a complex framework.** While the controlled and reasonable generation results are impressive, the overall system appears to be a loose combination of prior works like TrajectoryCrafter and IC-Light. The authors should better detail why jointly controlling both camera and illumination is an important task and which specific module Light-X introduces to better fit this combined task.

* **Need for direct geometry comparison and evaluation.** The paper claims that the relighted video "remains geometrically coherent," but no direct experiments are provided to support this crucial claim. A stronger evaluation would involve running the input and relighted videos through methods like VGGT (for static scenes) or CUT3R (for dynamic scenes) to obtain corresponding point clouds. The authors could then calculate the Chamfer distance between these paired point clouds and provide visualizations for better qualitative assessment.

* **Deeper analysis required for the global illumination control module.** The global illumination control module is introduced to prevent the diminishment of illumination effects. To illustrate its contribution more clearly, a visual comparison from an ablation study of this module should be provided. Furthermore, the authors are suggested to analyze the module's effect on regions with non-Lambertian surfaces. I am concerned that enforcing the same global light effects might negatively impact performance in regions that naturally reflect differently under various lighting conditions.

**Questions:**

Kindly refer to the [Weaknesses] section.

---

> ### Author Response · Authors · 2025-11-24
>
> We sincerely thank Reviewer 6GDF for the thoughtful review and constructive feedback. We address all concerns below.
>
> **W1.1: Why jointly controlling both camera and illumination is an important task?**
>
> Real-world appearance is jointly determined by geometry, motion, and illumination.
> Jointly controlling these factors enables more faithful 3D scene modeling, and potentially benefits applications such as AR/VR, film and content creation (multi-view and multi-illumination scene generation), as well as embodied AI and autonomous driving (realistic environment simulation for interaction).
>
> **W1.2: Which specific modules Light-X introduces for joint camera-illumination control?**
>
> 1) A loose combination of TrajectoryCrafter, IC-Light, or existing video relighting methods is insufficient, as shown by our quantitative (Tables 1 and 2) and qualitative comparisons (Fig.5, Fig.Q, and supplementary videos).
>
> 2) Light-X introduces components specifically designed for joint control: **i) Data pipeline:** a degradation-based pipeline with inverse geometric mapping across static, dynamic, and AI-generated sources, addressing the absence of real paired data. **ii) Problem formulation:** a disentangled conditioning scheme that explicitly separates geometry/motion cues from illumination, enabling both independent and coupled control. **iii) Model design:** fine-grained lighting cues, global illumination control, and a soft-mask mechanism, enabling coherent and robust lighting behavior across diverse illumination conditions.
>
> 3) **Ablation studies** confirm that each component is necessary, showing that Light-X is not a loose combination of prior works but a purpose-built framework for joint camera-illumination control.
>
> **W2: Geometry Consistency Evaluation.**
>
> We appreciate the reviewer’s insightful suggestion and performed point-cloud–based geometry evaluation using the state-of-the-art dynamic reconstruction method **MegaSAM** and the static reconstruction method **VGGT**.
> Point clouds reconstructed from both the input and relighted videos were compared using **Chamfer Distance (CD)**.
> The table below reports the CD metric, including the Mean, Median, Standard Deviation, Minimum, and Maximum values (* based on the first 16 frames). The results demonstrate the superior performance of our method, which yields the lowest mean and median distance.
> The corresponding point-cloud visualizations and video comparisons are provided in the supplementary material (**6GDF-W2-Point-Clouds.mp4**).
>
> | Method              | Avg CD ↓ | Median ↓ | Std ↓  | Min ↓  | Max ↓  |
> |---------------------|----------|----------|--------|--------|--------|
> | IC-Light            | 0.5012   | 0.1933   | 1.3300 | 0.0056 | 15.6113|
> | LAV                 | 0.8979   | 0.1903   | 4.5258 | 0.0096 | 58.0839|
> | Ours            | **0.3753** | **0.1581** | 0.7228 | 0.0063 | 5.9780 |
> | IC-Light + AnyV2V   | 0.8896   | 0.2813   | 1.8282 | 0.0055 | 19.9080|
> | Ours*           | **0.3784** | **0.1577** | 0.7535 | 0.0064 | 5.8883 |
>
>
> **W3.1: Ablation Visual Comparison of the Global Illumination Control Module.**
>
> Following the reviewer’s suggestion, we provide the visual comparison for the ablation of the global illumination control module in **Fig.P** of the revised manuscript, with the corresponding video results included in the supplementary material (**6GDF-W3.1-Global-Ablation.mp4**).
> This comparison more clearly demonstrates that the global illumination control module effectively prevents the attenuation and fading of illumination effects.
>
> **W3.2: Analysis on Non-Lambertian Surfaces.**
>
> The reviewer raises an interesting and valuable point concerning non-Lambertian regions.
> The global illumination control module generates only a coarse illumination signal by using illumination tokens to query the reference image, thereby avoiding the override of fine-grained details in non-Lambertian regions. The accuracy of specular and view-dependent effects is mainly supported by fine-grained lighting cues, real-data supervision, and the strong priors of the pre-trained video diffusion model. Additional relighting results on non-Lambertian regions are provided in **Fig.G** of the revised manuscript and in the supplementary material (**6GDF-W3.2-Non-Lambertian.mp4**).

---

### Author Response · Authors · 2025-11-27
**Summary of Responses**

We sincerely thank all reviewers for their efforts and constructive feedback. The pre-rebuttal ratings are **8 (Reviewer JNPV), 6 (Reviewer 6GDF), 6 (Reviewer LGn4), and 6 (Reviewer BDHW)**. We appreciate the reviewers’ recognition of the significance of our work and their insightful suggestions, which have helped further strengthen the paper.

---

### Updates Addressing Reviewer 6GDF

- **Comprehensive geometry validation via point-cloud comparison**
  • Quantitative results added in **Table D** in the revised manuscript
  • Visualization provided as **6GDF-W2-Point-Clouds.mp4** in the supplementary ZIP package

- **Ablation of the global illumination control module**
  • Visual comparison added in **Fig. P** in the revised manuscript
  • Corresponding video provided as **6GDF-W3.1-Global-Ablation.mp4** in the supplementary ZIP package

- **Performance on non-Lambertian surfaces**
  • Results added in **Fig. G** in the revised manuscript
  • Corresponding video provided as **6GDF-W3.2-Non-Lambertian.mp4** in the supplementary ZIP package

---

### Updates Addressing Reviewer JNPV

- **New real-video evaluation without IC-Light reference**
  • Results added in **Table 2** and **Table 4** in the revised manuscript

- **Large camera trajectories**
  • Results added in **Fig. H** in the revised manuscript
  • Corresponding video provided as **JNPV-Q1-Large-Camera.mp4** in the supplementary ZIP package

---

### Updates Addressing Reviewer LGn4

- **New real-video evaluation without IC-Light reference**
  • Results added in **Table 2** and **Table 4** in the revised manuscript

- **Extended comparisons with recent methods (ReCamMaster, Free4D, TC-Light)**
  • Results added in **Table E**, **Table F**, and **Table G** in the revised manuscript

- **Robustness to depth noise**
  • Results added in **Table H** and **Fig. K** in the revised manuscript
  • Corresponding video provided as **LGn4-Q1.3&Q2-Depth-Noise.mp4** in the supplementary ZIP package

- **FID degradation with temporal distance**
  • Curve added in **Fig. J** in the revised manuscript

- **Evaluation of different relighting reference-frame selection strategies**
  • Results added in **Table I** in the revised manuscript

- **Data source ablation (static / dynamic / AI-generated)**
  • Results added in **Fig. M**, **Fig. N**, and **Fig. O** in the revised manuscript
  • Corresponding video provided as **LGn4-W3-Data-Ablation.mp4** in the supplementary ZIP package

- **Performance with occluded relit frames**
  • Results added in **Fig. I** in the revised manuscript
  • Corresponding video provided as **LGn4-Q1.2-Reference-Occlusion.mp4** in the supplementary ZIP package

---

### Updates Addressing Reviewer BDHW

- **Using Light-X itself for data generation**
  • Results added in **Table J** in the revised manuscript

- **Use of synthetic relighting data**
  • Results added in **Table K** in the revised manuscript

---

All revisions are highlighted in **blue** in the revised manuscript, and all new **video results** are included in the **supplementary materials**. Point-by-point responses are provided in each reviewer discussion thread. We sincerely thank all reviewers again for their constructive suggestions and engagement throughout the review process.

---

### Meta-Review · Area_Chair_shPF · 2026-01-07

**Summary:**

The reviewers reached a strong consensus to accept this paper, praising "Light-X" for its novel approach to joint camera trajectory and illumination control in video generation. Key strengths identified include the disentangled design separating geometry from lighting, the effective "Light-Syn" data curation pipeline which synthesizes training pairs from in-the-wild footage, and the impressive empirical results that surpass existing two-stage baselines. The rebuttal was highly effective, providing extensive additional experiments which resolved the reviewers' primary technical concerns.

**Reviewer Concerns:**

Reviewer Concerns addressed by the rebuttal:

Evaluation Metric Bias (Reviewers JNPV, LGn4): The concern that using IC-Light-based FID favors the method was addressed by a new "real-video degradation" experiment using standard metrics (PSNR, SSIM, LPIPS, FVD) against ground truth videos, where Light-X consistently outperformed baselines.

Insufficient Baselines (Reviewer LGn4): The authors added quantitative comparisons against recent state-of-the-art methods (ReCamMaster, Free4D, TC-Light), demonstrating superior performance in both joint control and individual tasks.

Geometry Consistency Validation (Reviewer 6GDF): The claim of geometric coherence was substantiated with a new quantitative evaluation using Chamfer Distance on reconstructed point clouds.


Reviewer Concerns that remains:

Optimal Reference Frame Selection (Reviewer LGn4): While the method is robust to random reference frame selection, the authors acknowledged that automatically selecting the most informative frame is an interesting direction left for future work.

**Reviewer Scores:**

The reviewers will all maintain their positive scores.

---

### Decision · Program_Chairs · 2026-01-26

Accept (Poster)